# A *MSTN*^Del73C mutation with *FGF5* knockout sheep by CRISPR/Cas9 promotes skeletal muscle myofiber hyperplasia

Ming-Ming Chen[1†], Yue Zhao[1†], Kun Yu[1†], Xue-Ling Xu[1], Xiao-Sheng Zhang[2], Jin-Long Zhang[2], Su-Jun Wu[1], Zhi-Mei Liu[1], Yi-Ming Yuan[1], Xiao-Fei Guo[2], Shi-Yu Qi[1], Guang Yi[1], Shu-Qi Wang[1], Huang-Xiang Li[1], Ao-Wu Wu[1], Guo-Shi Liu[1], Shou-Long Deng[3], Hong-Bing Han[1], Feng-Hua Lv[1*], Di Lian[4*], Zheng-Xing Lian[1*]

[1]State Key Laboratory of Animal Biotech Breeding, Beijing Key Laboratory for Animal Genetic Improvement, National Engineering Laboratory for Animal Breeding, Key Laboratory of Animal Genetics and Breeding of the Ministry of Agriculture, College of Animal Science and Technology, China Agricultural University, Beijing, China; [2]Institute of Animal Husbandry and Veterinary Medicine, Tianjin Academy of Agricultural Sciences, Tianjin, China; [3]National Center of Technology Innovation for animal model, NHC Key Laboratory of Human Disease Comparative Medicine, Institute of Laboratory Animal Sciences, Chinese Academy of Medical Sciences and Comparative Medicine Center, Peking Union Medical College, Beijing, China; [4]College of Pulmonary and Critical Care Medicine, Chinese PLA General Hospital, Beijing, China

*For correspondence:
lvfenghua@cau.edu.cn (F-HL);
b20173020099@cau.edu.cn (DL);
lianzhx@cau.edu.cn (Z-XL)

†These authors contributed equally to this work

**Abstract** Mutations in the well-known Myostatin (*MSTN*) produce a 'double-muscle' phenotype, which makes it commercially invaluable for improving livestock meat production and providing high-quality protein for humans. However, mutations at different loci of the *MSTN* often produce a variety of different phenotypes. In the current study, we increased the delivery ratio of Cas9 mRNA to sgRNA from the traditional 1:2 to 1:10, which improves the efficiency of the homozygous mutation of biallelic gene. Here, a *MSTN*^Del73C mutation with *FGF5* knockout sheep, in which the *MSTN* and *FGF5* dual-gene biallelic homozygous mutations were produced via the deletion of 3-base pairs of AGC in the third exon of *MSTN*, resulting in cysteine-depleted at amino acid position 73, and the *FGF5* double allele mutation led to inactivation of *FGF5* gene. The *MSTN*^Del73C mutation with *FGF5* knockout sheep highlights a dominant 'double-muscle' phenotype, which can be stably inherited. Both F0 and F1 generation mutants highlight the excellent trait of high-yield meat with a smaller cross-sectional area and higher number of muscle fibers per unit area. Mechanistically, the *MSTN*^Del73C mutation with *FGF5* knockout mediated the activation of *FOSL1* via the MEK-ERK-FOSL1 axis. The activated *FOSL1* promotes skeletal muscle satellite cell proliferation and inhibits myogenic differentiation by inhibiting the expression of MyoD1, and resulting in smaller myotubes. In addition, activated ERK1/2 may inhibit the secondary fusion of myotubes by $Ca^{2+}$-dependent CaMKII activation pathway, leading to myoblasts fusion to form smaller myotubes.

## eLife assessment

The authors present a **useful** analysis of the phenotype of sheep in which the muscle developmental regulator myostatin has been mutated in a FGF5 knockout background. The goal was to produce

sheep with a "double-muscled" phenotype, yet the genetically engineered sheep exhibited meat with a smaller cross-sectional area and higher number of muscle fibers. The work extends the extensive body of knowledge already published in this area. The authors provide evidence using in vitro experiments that Fosl1 regulates myogenesis, but the strength of evidence relating to the muscle phenotype and underlying cellular and molecular mechanism remains **incomplete**.

## Introduction

Myostatin (*MSTN*) has been well-known as a negative regulator of muscle growth and development. Its mutation produces a 'double-muscle' phenotype, which shows its inestimable commercial value in improving meat production of livestock and poultry, and providing high-quality protein for humans (*Fan et al., 2022*; *Chen et al., 2021b*). Due to its role in promoting muscle atrophy and cachexia, *MSTN* has been recognized as a promising therapeutic target to offset the loss of muscle mass (*Lee, 2021*; *Baig et al., 2022*; *Wijaya et al., 2022*).

*MSTN* is highly conserved in mammals, and mutations in the *MSTN* gene, either artificially or naturally, will result in increased skeletal muscle weight and produce a 'double-muscle' phenotype, which has been reported in many species, including cattle, sheep, and pigs, rabbits, and humans (*Grisolia et al., 2009*; *Dilger et al., 2010*; *Kambadur et al., 1997*). However, mutations at different loci of the *MSTN* often produce variety of different phenotypes, and its molecular mechanism of skeletal muscle growth and development remains controversial (*Hanset and Michaux, 1985*; *Grobet et al., 1997*; *Wegner et al., 2000*; *Kambadur et al., 1997*; *Marchitelli et al., 2003*). More than 77 natural mutation sites of *MSTN* have been reported in various sheep breeds, most of these mutations were found to be located in the non-coding regions, and did not affect *MSTN* activity (*Kijas et al., 2007*; *Sjakste et al., 2011*; *Han et al., 2013*; *Dehnavi et al., 2012*). In addition to introns, it is still possible that mutations in regulatory regions and exons may not affect the sheep phenotypes (*Pothuraju et al., 2015*; *Kijas et al., 2007*; *Boman and Våge, 2009*; *Boman et al., 2009*).

Fibroblast growth factor 5 (*FGF5*) belongs to the fibroblast growth factor (FGF) family and is a secretory signaling protein. *FGF5* played an inhibitory effect on mouse hair growth (*Hébert et al., 1994*), and its natural mutation can lead to a significant increase in hair growth in angora mice (*Sundberg et al., 1997*). Subsequent studies have also successively confirmed the inhibitory effect of *FGF5* on mammalian hair growth and is recognized to be a negative regulator of hair growth (*Kehler et al., 2007*; *Dierks et al., 2013*; *Yoshizawa et al., 2015*; *Legrand et al., 2014*; *Higgins et al., 2014*).

In this study, to increase both meat and wool production, we first produced the *MSTN* and *FGF5* dual-gene biallelic homozygous mutations sheep by the increased delivery ratio of Cas9 mRNA to sgRNA targeting *MSTN* and *FGF5*. The *MSTN*^Del73C mutation with *FGF5* knockout sheep highlights a dominant 'double-muscle' phenotype by decreasing the muscle fiber cross-sectional area and increasing the number of muscle fibers per unit area. Then, we used the *MSTN* and *FGF5* dual-gene biallelic homozygous mutations sheep to unravel the molecular mechanism of the 'double-muscle' phenotype and myofiber hyperplasia.

## Results

### Elevated molar ratio of Cas9/sgRNA can efficiently generate biallelic homozygous mutant sheep

The sgRNAs for targeting were designed in the third exon of the MSTN and FGF5 genes, respectively (*Figure 1A and B*). Both MSTN and FGF5 PCR products could be cleaved by T7E1 and the fragment sizes were also as expected, and grayscale analysis showed that the editing efficiency was 14.6% and 11.4%, respectively (*Figure 1C*), which indicates that the designed sgRNAs can achieve more efficient gene targeting. The microinjection was performed according to the molar ratio of Cas9 mRNA:sgRNAs (1:2, 1:10, and 1:15), respectively. The number of embryos injected, recipients of nuclear transfer, pregnancy, and alive lambs per group were listed in *Supplementary file 1A*. The subsequent gene mutation detection showed that a total of 3 lambs were mutated in the MSTN and FGF5 genes at a Cas9 mRNA:sgRNAs injection molar ratio of 1:2, with a gene editing mutation rate of 14.3% (4/28). However, all three lambs were chimeric, that is, there were both mutant and

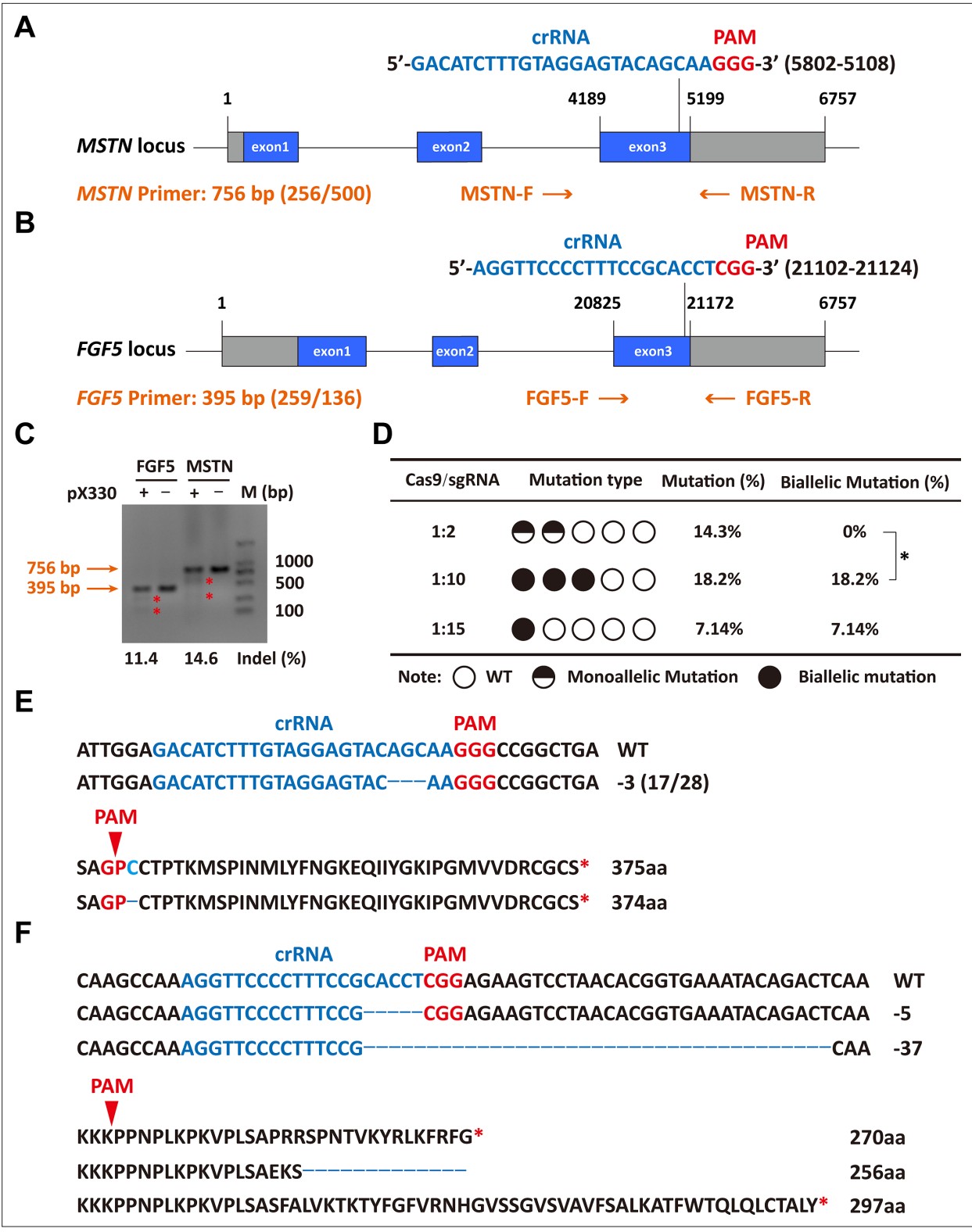

**Figure 1.** Efficient generation of sheep carrying biallelic mutations in dual gene via the CRISPR/Cas9 system. (**A**) Schematic of sgRNAs specific to exon 3 of the sheep MSTN locus. The crRNA sequences are highlighted in blue typeface and the PAM in red. (**B**) Schematic of sgRNAs specific to exon 3 of the sheep FGF5 locus. The crRNA sequences are highlighted in blue typeface and the PAM in red. (**C**) T7EI assay for sgRNAs of MSTN and FGF5 in sheep fetal fibroblasts. The cleavage bands are marked with an red asterisk (*) and the indel frequencies were calculated using the expected fragments. (**D**) Summary of the generation of sheep carrying biallelic mutations in dual gene via zygote injection of Cas9 mRNA/sgRNAs. Biallelic mutation rate was

*Figure 1 continued on next page*

*Figure 1 continued*

statistically analyzed using chi square test. *p<0.05. (**E**) Analysis of genome sequence and amino acid sequence of MSTN-modified sheep. The location of sgRNA and PAM are highlighted in blue and red, respectively. The deletions are indicated by a dashed line (-). (**F**) Analysis of genome sequence and amino acid sequence of FGF5-modified sheep. The location of sgRNA and PAM are highlighted in blue and red, respectively. The deletions are indicated by a dashed line (-).

The online version of this article includes the following source data for figure 1:

**Source data 1.** Uncropped and labeled gels for *Figure 1C*.

**Source data 2.** Raw unedited gels for *Figure 1C*.

wild-type after editing, and the biallelic mutation rate was 0% (*Supplementary file 1A*, *Figure 1D*). Increasing the injection molar ratio of Cas9 mRNA:sgRNAs to 1:10 resulted in mutations in MSTN and FGF5 genes of two lambs with a gene editing mutation rate of 18.2% (4/22), which contributed to a significant (p<0.05) increase in the biallelic mutation rate (*Supplementary file 1A*, *Figure 1D*). While the injection molar ratio of Cas9 mRNA: sgRNAs was continuously increased to 1:15, one lamb had a mutation, which was a biallelic mutation of *MSTN* gene, and the gene editing mutation rate was 7.14% (1/14; *Supplementary file 1A*, *Figure 1D*). These results indicate that increasing the delivery molar ratio of Cas9 mRNA to sgRNA from 1:2 to 1:10 can greatly improve the efficiency of biallelic mutation in sheep.

## The $MSTN^{Del73C}$ mutation with *FGF5* knockout sheep highlights a dominant "double-muscle" phenotype and muscle fiber hyperplasia

Among gene-edited sheep, a sheep with biallelic deletion of *MSTN* and biallelic mutation of *FGF5* aroused our great interest. Specifically, gene editing caused a deletion of 3-base pairs of AGC in the third exon of *MSTN* (*Figure 1E and F*), resulting in the deletion of cysteine at amino acid position 73 ($MSTN^{Del73C}$) of the mature peptide (*Figure 1E and F*), which is highlighted by the 'double-muscle' phenotype (*Figure 2A and B*). At the same time, a biallelic mutation in *FGF5* caused the knockout of *FGF5* gene and increased the density and length of hairs (*Zhang et al., 2020*). Compared to WT sheep, the fiber cell number per unit area of gluteus medius and longissimus dorsi in $MF^{-/-}$ sheep was significantly (p<0.01) increased (*Figure 2C and D*), and the cross-sectional area were smaller (*Figure 2C and E–F*). Similarly, the cross-sectional area of gluteus medius muscle fibers in the offspring generation $MF^{+/-}$ sheep was also smaller (*Figure 2G–H*), and the number of muscle fiber cells per unit area was significantly increased (p<0.0001) (*Figure 2G,I*); the percentage of smaller muscle fiber area in $MF^{+/-}$ sheep was significantly increased (p<0.05) (*Figure 2G and J*), these results was consistent with that in $MF^{-/-}$ sheep. Interestingly, the FGF5 knockout alone had no significant (p>0.05) effect on muscle fiber size (*Figure 2—figure supplement 1A–D*). Although the mRNA expression levels of *MSTN* and *FGF5* were significantly (p<0.05) reduced in $MF^{+/-}$ sheep during early embryonic development (3-month-old; *Figure 2—figure supplement 2A*), there was no significant differences in *MSTN* mRNA and protein expression levels at 12-month-old after birth (*Figure 2—figure supplement 2B–D*). In addition, there was no significant differenc in the proportion of centrally nucleated myofibres (p>0.05; *Figure 2—figure supplement 2E*), nor were there aberrant expression of some genes related to muscular dystrophy and muscle atrophy (*Figure 2—figure supplement 2F*). These results indicate that $MSTN^{Del73C}$ mutation with *FGF5* knockout produces muscle fiber hyperplasia instead of muscular dystrophy or muscle atrophy.

Further, although the muscle weight of different parts in WT and $MF^{+/-}$ sheep has no significant difference (*Supplementary file 1B*), the proportion of hind leg meat was significantly (p<0.05) increased by 21.2% (*Supplementary file 1C*), and the proportion of gluteus medius in the carcass of $MF^{+/-}$ sheep was significantly (p<0.01) increased by 26.3% compared to WT sheep (*Figure 2K*). In addition, there were no significant (p>0.05) differences in pH, color, drip loss, cooking loss, shearing force, and amino acid content of the longissimus dorsi between WT and $MF^{+/-}$ sheep (*Supplementary file 1D-F*). All these results demonstrated that the $MSTN^{Del73C}$ mutation with *FGF5* knockout sheep had well-developed hip muscles with smaller muscle fibers, which do not affect meat quality, and this phenotype may be dominated by *MSTN* gene.

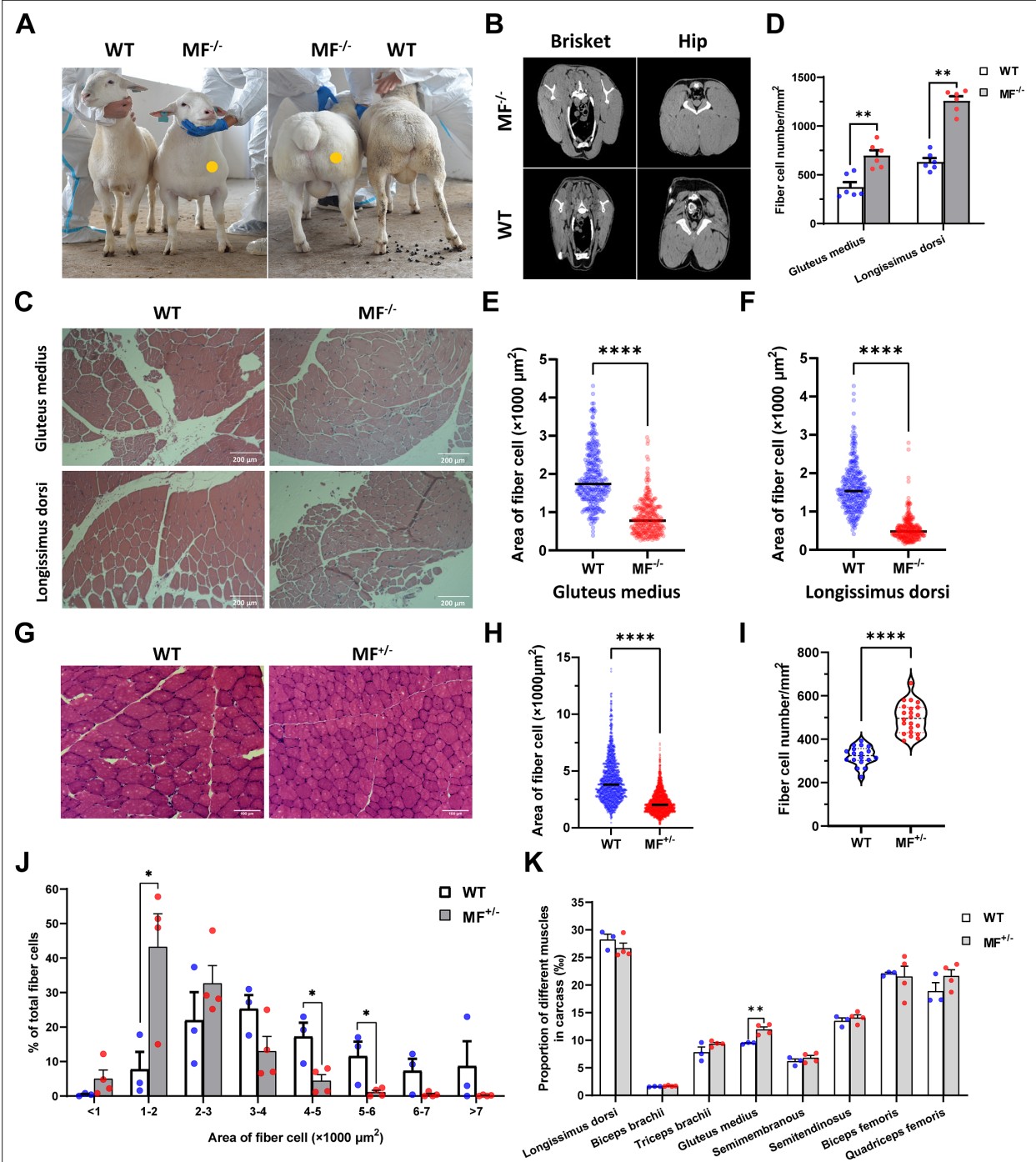

**Figure 2.** The *MSTN*^Del73C mutation with *FGF5* knockout sheep highlights a dominant 'double-muscle' phenotype and muscle fiber hyperplasia. (**A**) The 6-month-old WT and MF^-/- sheep. The genome-edited sheep displayed an obvious 'double-muscle' phenotype compared with the WT. (**B**) The CT scanning image of the brisket and hip of WT and MF^-/- sheep. (**C**) HE sections of gluteus medius and longissimus dorsi of WT and MF^-/- sheep. Scale bar 200 μm. (**D**) Quantification of muscle fibre cell number of per unit area in WT (n=3) and MF^-/- (n=1) sheep. All data points were shown. (**E–F**) Quantification of muscle fibre cell area of gluteus medius and longissimus dorsi in WT (n=3) and MF^-/- (n=1) sheep. All data points were shown. (**G**) HE sections of gluteus medius in WT and MF^+/- sheep. Scale bar 100 μm. (**H**) Quantification of muscle fibre cell area of gluteus medius in WT (n=3) and MF^+/- (n=4) sheep. (**I**) Quantification of muscle fibre cell number of per unit area in WT (n=3) and MF^+/- (n=4) sheep. (**J**) The percentage of cross-sectional area of different size muscle fibers. (**K**) The proportion of different muscles in carcass in WT (n=3) and MF^+/- (n=4) sheep. Data: mean ± SEM. Unpaired student's t-test were used for statistical analysis after the equal variance test, otherwise the t-test with Welch's correction were performed. *p<0.05, **p<0.01, ***p<0.001, and ****p<0.0001.

*Figure 2 continued on next page*

*Figure 2 continued*

The online version of this article includes the following source data and figure supplement(s) for figure 2:

**Figure supplement 1.** *FGF5* mutation does not affect muscle fiber size.

**Figure supplement 2.** The *MSTN*<sup>Del73C</sup> mutation with *FGF5* knockout has no potential effect on MSTN expression and muscular dystrophy.

**Figure supplement 2—source data 1.** Uncropped and labeled blots for *Figure 2—figure supplement 2C*.

**Figure supplement 2—source data 2.** Raw unedited blots for *Figure 2—figure supplement 2C*.

## The *MSTN*$^{Del73C}$ mutation with *FGF5* knockout promotes skeletal muscle satellite cells proliferation and inhibits myogenic differentiation

The proliferation and differentiation of skeletal muscle satellite cells is a key step in muscle formation and development. The CCK-8 and EdU cell proliferation experiments showed that the proliferative rate of MF$^{+/-}$ cells were highly significantly ($p<0.01$) elevated (*Figure 3A*) with a significant ($p<0.05$) increase in the rate of EdU-positive cells (*Figure 3B and C*) compared to WT cells. In addition, cell cycle detection showed a significant ($p<0.01$) reduce in the proportion of G1 phase and a significant increase ($p<0.05$) in the proportion of S phase in MF$^{+/-}$ cells (*Figure 3D and E*). Meanwhile, the mRNA expression levels of the cell cycle marker genes CyclinB1, CDK4, Cyclin A1, Cyclin E1, and CDK2 were significantly increased ($p<0.05$; *Figure 3F*). These results suggest that the *MSTN*$^{Del73C}$ mutation with *FGF5* knockout may promote cell proliferation by accelerating the cell cycle from G0/G1 phase to S phase.

The mRNA level of MyHC and the protein levels of MyoD1, MyoG, and MyHC (*Figure 3G–I*) were dramatically decreased ($p<0.05$) after induced differentiation 2 days in MF$^{+/-}$ cells, suggesting that the *MSTN*$^{Del73C}$ mutation with *FGF5* inhibit myogenic differentiation. Meanwhile, the immunofluorescence staining of MyoG and MyHC in myotubes showed that myotube fusion index (*Figure 3J and K*), number of myotubes (*Figure 3J and L*), and number of nuclei per myotube (*Figure 3J and M*) were all highly significantly ($P<0.01$) reduced after inducing differentiation for 2 days of MF$^{+/-}$ cells compared to WT cells, as was the myotube diameter at the maximum measured (*Figure 3J and N*). In addtion, the differentiation capacity and fusion ability of MF$^{+/-}$ cells were consistently significantly lower than WT cells during the ongoing differentiation process, as was the diameter of fused myotubes (*Figure 3—figure supplement 1A–J*). The reduced expression of myogenic differentiation markers further confirmed that the *MSTN*$^{Del73C}$ mutation with *FGF5* knockout consistently inhibits myogenic differentiation of skeletal muscle satellite cells (*Figure 3—figure supplement 1K–N*). Taken together, our results elucidated that the *MSTN*$^{Del73C}$ mutation with *FGF5* knockout promotes proliferation and inhibits myogenic differentiation of skeletal muscle satellite cells, and induces a smaller myotube diameter of myotubes, which may explain the muscle fiber hyperplasia phenotype and the decreased cross-sectional area of muscle fibers in MF$^{-/-}$ and MF$^{+/-}$ sheep.

## The *MSTN*$^{Del73C}$ mutation with *FGF5* knockout contribute to muscle phenotype via MEK-ERK-FOSL1 axis

To elucidate the potential mechanism of the *MSTN*$^{Del73C}$ mutation with *FGF5* knockout result in smaller muscle fiber cross-sectional area and myotube diameter, the RNA-seq was performed in gluteus medius. GO and KEGG enrichment analysis indicating that differentially expressed genes (DEGs) were significantly closely related to cell proliferation, myogenic differentiation, and muscle development; and significantly enriched in MAPK signaling pathway (*Figure 4A–B*). Among DEGs, FOSL1 has aroused our interest. Compared to WT sheep, the FSOL1 mRNA level was significantly ($p<0.05$) lower in both gluteus medius and longissimus dorsi in MF$^{+/-}$ sheep (*Figure 4C*), but it was significantly ($p<0.05$) elevated in MF$^{+/-}$ cells at GM and DM2 (*Figure 4D*). More strikingly, FOSL1 mRNA level was strongly ($p<0.01$) decreased after induced differentiation (*Figure 4E*), and its expression diminished continuously with the differentiation progress. These results suggested that FOSL1 may play a crucial role in the proliferation and myogenic differentiation of skeletal muscle satellite cells.

FOSL1 is a member of the AP-1 family, and another member of this family, c-Fos, inhibits myogenesis and MyoD1 expression by directly binding to the MyoD1 promoter region. Therefore, we speculated that FOSL1 might have similar functions to c-Fos. Subsequent protein-protein interaction (PPI) analysis further suggested that there was a potential interaction between FOSL1 and MyoD1 (*Figure 4F*). In

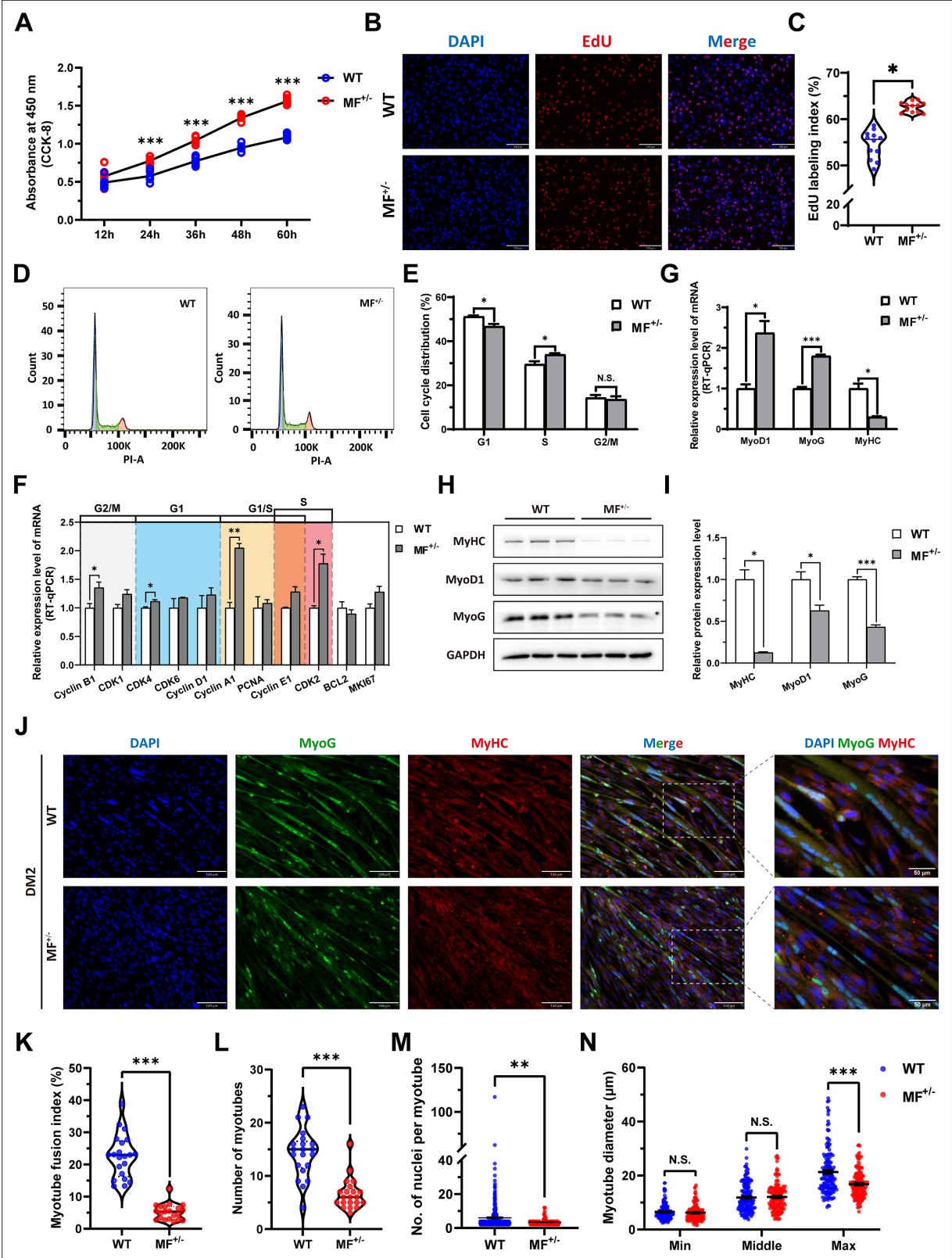

**Figure 3.** The *MSTN*<sup>Del73C</sup> mutation with *FGF5* knockout promote proliferation and inhibit differentiation of skeletal muscle satellite cells. (**A**) The number of cells was detected by CCK-8 at 12 hr, 24 hr, 36 hr, 48 hr, and 60 hr in GM (n=7–8 per group). (**B–C**) EdU assay showed that the number of EdU positive cells and EdU labeling index were significantly increased in MF⁺/⁻ cells (n=3). Scale bar 130 μm. All data points were shown. (**D–E**) PI staining to detect cell cycle and showed a significant reduce in the proportion of G1 phase and a significant increase in the proportion of S phase in MF⁺/⁻ cells (n=4).

*Figure 3 continued on next page*

*Figure 3 continued*

(**F**) The mRNA expression levels of cell cycle marker genes and cell proliferation marker genes (n=3). (**G**) The mRNA expression levels of myogenic differentiation marker genes MyoG, MyoD1, and MyHC (n=3). (**H–I**) The protein expression levels of myogenic differentiation marker genes MyoG, MyoD1, and MyHC (n=3). (**J**) The MyoG and MyHC immunofluorescence staining of myotubes in DM2. Scale bar 130 μm. (**K**) The myotube fusion index, which was represented by the number of cell nuclei in myotubes/total cell nuclei (n=3). All data points were shown. (**L**) The number of myotubes, which was the number of all myotubes in the field of view (n=3). All data points were shown. (**M**) The number of nuclei per myotube (n=3). All data points were shown. (**N**) The myotube diameter (n=3). To reflect the myotube diameter as accurately as possible, the vertical line at the thinnest position of the myotube is taken as the minimum measured (Min), the mid-perpendicular line of the long myotube axis is taken as the middle measured (Middle), and the vertical line at the widest position of the myotube is taken as the maximum measured (Max). All data points were shown. Data: mean ± SEM. Unpaired student's t-test and chi square test were used for statistical analysis. All student's t-test were performed after the equal variance test, otherwise the t-test with Welch's correction were used. *$p < 0.05$, **$p < 0.01$, and ***$p < 0.001$.

The online version of this article includes the following source data and figure supplement(s) for figure 3:

**Source data 1.** Uncropped and labeled blots for *Figure 3H*.

**Source data 2.** Raw unedited blots for *Figure 3H*.

**Figure supplement 1.** The myogenic differentiation ability of MF[+/-] cells was continuously inhibited.

**Figure supplement 1—source data 1.** Uncropped and labeled blots for *Figure 3—figure supplement 1K*.

**Figure supplement 1—source data 2.** Raw unedited blots for *Figure 3—figure supplement 1K*.

addition, the c-Fos mRNA level was highly significantly ($p < 0.01$) reduced in MF[+/-] myoblasts compared to WT cells, whereas the MyoD1 mRNA level was dramatically ($p < 0.01$) increased (*Figure 4G*). We futher found that two bZIP recognition sites in the MyoD1 promoter region had the most significant binding potential to FOSL1 (*Figure 4H–J*). Subsequently, ChIP-qPCR confirmed that FOSL1 directly binds to these two bZIP recognition sites in the MyoD1 promoter region (*Figure 4K–L*). The dual luciferase report experiment confirmed that transcription factor FOSL1 can significantly ($p < 0.05$) inhibit MyoD1 promoter activity (*Figure 4M*). These results indicated that FOSL1 plays an important role in the transcriptional regulation of MyoD1.

As mentioned above, FOSL1 may be involved in the proliferation and myogenic differentiation of skeletal muscle satellite cells. Here, the protein levels of FOSL1 and c-Fos were significantly ($p < 0.05$) reduced in MF[+/-] cells at GM compared to WT cells (*Figure 4N–O*), and accompanied by a significant ($p < 0.05$) increase in p-FOSL1 protein levels (*Figure 4N–O*), whereas FOSL1 protein levels were significantly ($p < 0.05$) diminished and c-Fos protein levels were highly significantly ($p < 0.01$) elevated in MF[+/-] cells after induced differentiation (*Figure 4P–Q*), these results further support the key role of FOSL1 on myogenesis. As demonstrated previously, enrichment analysis significantly enriched the MAPK signaling pathway. Compared to WT cells, ERK1/2 protein level was extremely significantly ($p < 0.01$) decreased, and accompanied by a significant ($p < 0.05$) increase in p-ERK1/2 protein levels (*Figure 4N–O*). After induced differentiation, although both MEK1/2 and ERK1/2 protein levels were dramatically ($p < 0.01$) inhibited (*Figure 4P–Q*).

Considering the possible serum regulation of MSTN, we examined the effects of MSTN mutations on its receptors and downstream target genes, and observed that both MSTN receptors were significantly up-regulated (*Figure 4—figure supplement 1A*), whereas the expression of downstream Smand and Jun families was also inhibited to a varying degree (*Figure 4—figure supplement 1B–C*). Furthermore, serum from MF[+/-] sheep promoted the proliferation of skeletal muscle satellite cells (*Figure 4—figure supplement 1D*). *MSTN*[Del73C] mutation with *FGF5* knockout promoted FOSL1 expression using WT sheep serum (*Figure 4—figure supplement 1E*), which was similar to the results of FBS culture and HS induction. The serum from MF[+/-] sheep strongly stimulated FOSL1 expression and the inhibition of MyoD1 (*Figure 4—figure supplement 1F*). These results suggest that serum regulation cannot be ignored after *MSTN*[Del73C] mutation with *FGF5* knockout.

In summary, *MSTN*[Del73C] mutation with *FGF5* knockout may regulate the expression and activity of FOSL1 via the MEK1/2-ERK1/2-FOSL1 axis to affect the proliferation and myogenic differentiation of skeletal muscle satellite cells, and further contribute to muscle phenotype.

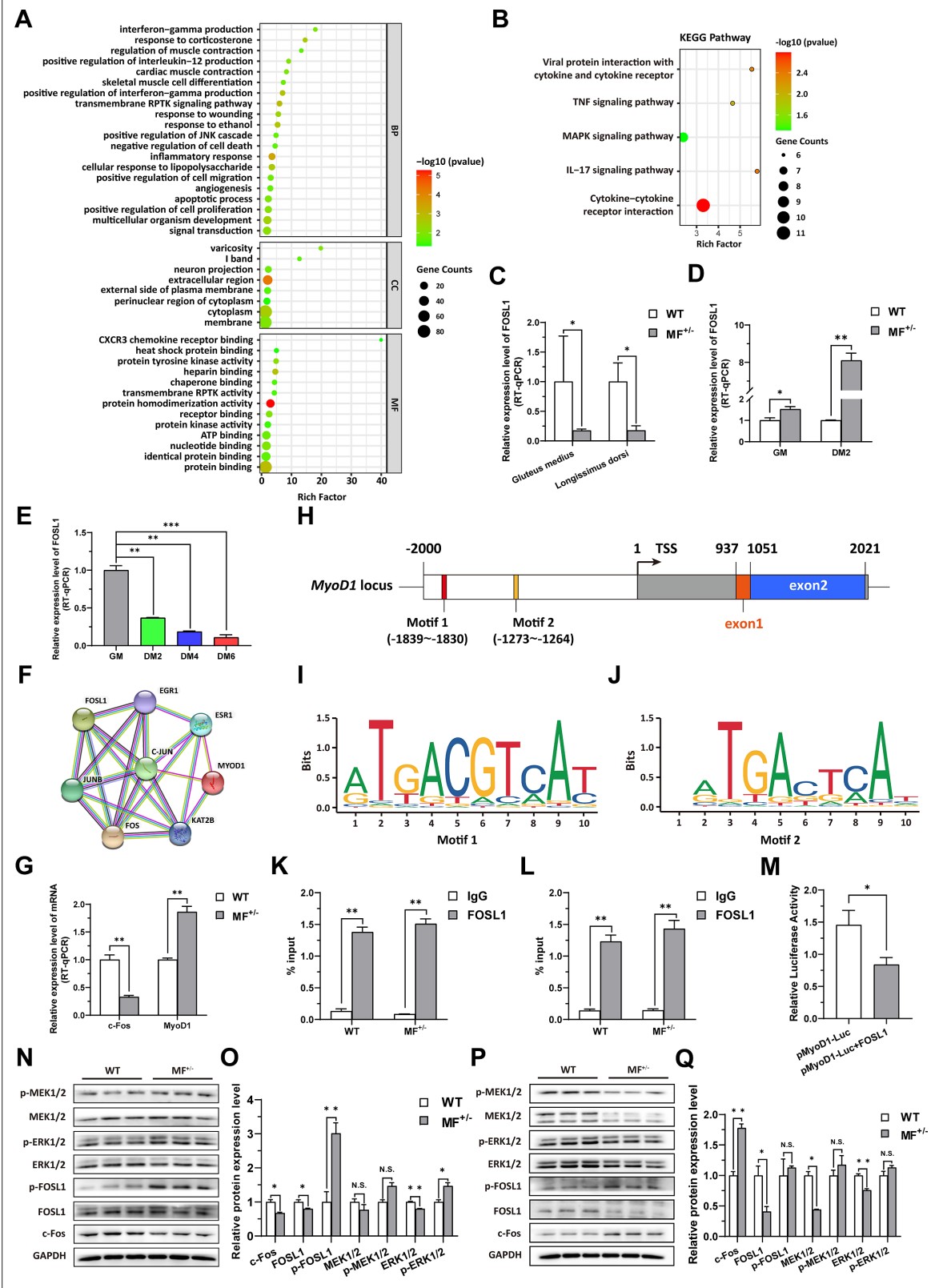

**Figure 4.** The *MSTN*[Del73C] mutation with *FGF5* knockout contributes to muscle phenotype via MEK-ERK-FOSL1 axis. (**A**) Go enrichment analysis of DEGs. Among them, the top 20 entries with significant enrichment are listed in biological process (BP). CC, cellular component; MF, molecular function. (**B**) KEGG enrichment analysis of DEGs. (**C**) The mRNA expression level of FOSL1 both at gluteus medius and longissimus dorsi in WT (n=3) and MF[+/-] (n=4) sheep. (**D**) The mRNA expression level of FOSL1 both at GM and DM2 in WT and MF[+/-] cells (n=3). (**E**) The expression level of FOSL1 mRNA

*Figure 4 continued*

during myogenic differentiation (n=3). (**F**) The protein-protein interaction (PPI) analysis of FOSL1, c-Fos and MyoD1. (**G**) The mRNA expression level of c-Fos and MyoD1 at GM in WT and MF$^{+/-}$ myoblasts (n=3). (**H**) Schematic diagram of MyoD1 gene body, promoter region and binding sites. (**I–J**) FOSL1 recognition motif in the MyoD1 promoter region. (**K**) FOSL1 ChIP-qPCR of motif 1 recognition region (n=3). (**L**) FOSL1 ChIP-qPCR of motif 2 recognition region (n=3). (**M**) Dual luciferase assay for the effect of FOSL1 on MyoD1 promoter activity (n=4). (**N**) Western blot of FOSL1, c-Fos, and key kinases of MAPK signaling pathways at GM. (**O**) Quantification of protein expression of FOSL1, c-Fos, and key kinases of MAPK signaling pathways at GM (n=3). (**P**) Western blot of FOSL1, c-Fos, and key kinases of MAPK signaling pathways at DM2. (**Q**) Quantification of protein expression of FOSL1, c-Fos, and key kinases of MAPK signaling pathways at DM2 (n=3). Data: mean ± SEM. Unpaired student's t-test was used for statistical analysis. All student's t-test were performed after the equal variance test, otherwise the t-test with Welch's correction were used. *p<0.05, **p<0.01, and ***p<0.001.

The online version of this article includes the following source data and figure supplement(s) for figure 4:

**Source data 1.** Uncropped and labeled blots for *Figure 4N* and *Figure 4P*.

**Source data 2.** Raw unedited blots for *Figure 4N* and *Figure 4P*.

**Figure supplement 1.** The effects of MSTN signaling pathway and MF$^{+/-}$ serum on proliferation and differentiation of skeletal muscle satellite cells.

## FOSL1 expression and activity control the proliferation and myogenic differentiation of skeletal muscle satellite cells

To investigate the role of FOSL1 on the proliferation and myogenic differentiation of skeletal muscle satellite cells, we successfully constructed *FOSL1* gain-of-function model (*Figure 5A*). We found that overexpression of *FOSL1* significantly (p<0.05) promotes cell proliferation (*Figure 5B–E*), suppressed c-Fos mRNA level (*Figure 5F*) (p<0.05), and inhibited MyoD1 mRNA (p<0.01) and protein (p<0.05) levels (*Figure 5F–H*). These results are consistent with what we observed in MF$^{+/-}$ cells at GM, suggesting a potential inhibitory effect of FOSL1 protein on MyoD1. In addition, the protein levels of MyoD1, MyoG and MyHC were all significantly decreased (p<0.05), proving that the myogenic differentiation was inhibited after FOSL1 was overexpressed (*Figure 5I–J*). Subsequently, immunofluorescence staining further confirmed the significant (p<0.05) inhibitory effect on cell differentiation (*Figure 5K–L*). Also, the number of myotubes, the number of nuclei per myotube, and the myotube diameter were all significantly decreased (p<0.05) (*Figure 5K and M–O*). Complementarily, we also constructed *FOSL1* loss-of-function model (*Figure 6A*). In contrast to what observed in FOSL1 over-expression, interference with FOSL1 inhibited skeletal muscle satellite cell proliferation (*Figure 6B–E*), elevated c-Fos and MyoD1 mRNA levels (*Figure 6F*) and significantly (p<0.05) increased MyoD1 protein level (*Figure 6G–H*). With myogenic differentiation markers were significantly up-regulated (*Figure 6I–J*), the myotube fusion index, the number of myotubes, the number of nuclei per myotube, and the myotube diameter were all significantly increased (*Figure 6K–O*). These results further demonstrated that elevated FOSL1 level inhibits myogenic differentiation and produces smaller myotubes.

To further ascertain this insight, the tert-butylhydroquinone (TBHQ), which can strongly activate ERK1/2 and increase p-ERK1/2 protein expression level, was used to activate ERK1/2 and act as an indirect activator of FOSL1. As expected, the addition of 20 μM TBHQ significantly (p<0.01) inhibited the myogenic differentiation of skeletal muscle satellite cells (*Figure 7A–B*). And, the number of myotubes was significantly increased (p<0.05) (*Figure 7A and C*), while the number of nuclei per myotube was significantly (p<0.05) decreased and produced a smaller myotube diameter (p<0.05) (*Figure 7A and D–E*). In addition, PD98509, which can inhibit ERK1/2 and dose-dependent reduce p-ERK1/2 protein expression level, was used for complementary experiments. We observed opposite results to TBHQ treatment, including increase in myotube fusion index, the number of nuclei per myotube, and the myotube diameter (*Figure 7F–J*). Taken together, our results shed light that the *MSTN*$^{Del73C}$ mutation with *FGF5* knockout mediated the activation of FOSL1 via MEK-ERK-FOSL1 axis, further promotes skeletal muscle satellite cell proliferation, and inhibits myogenic differentiation by inhibiting the transcription of MyoD1, and resulting in smaller myotubes.

## The *MSTN*$^{Del73C}$ mutation with *FGF5* knockout inhibit calcium-dependent transcription signal pathway to regulate secondary fusion of myotubes

As mentioned previously, DEGs identified by RNA-seq were significantly enriched in biological processes such as muscle contraction and cardiac muscle contraction. In muscle contraction, calcium ions (Ca$^{2+}$) play a crucial role, triggering the process of muscle contraction and relaxation by binding to proteins in muscle fibers. Here, the calcium-calmodulin dependent protein kinase II (CaMKII) α/δ

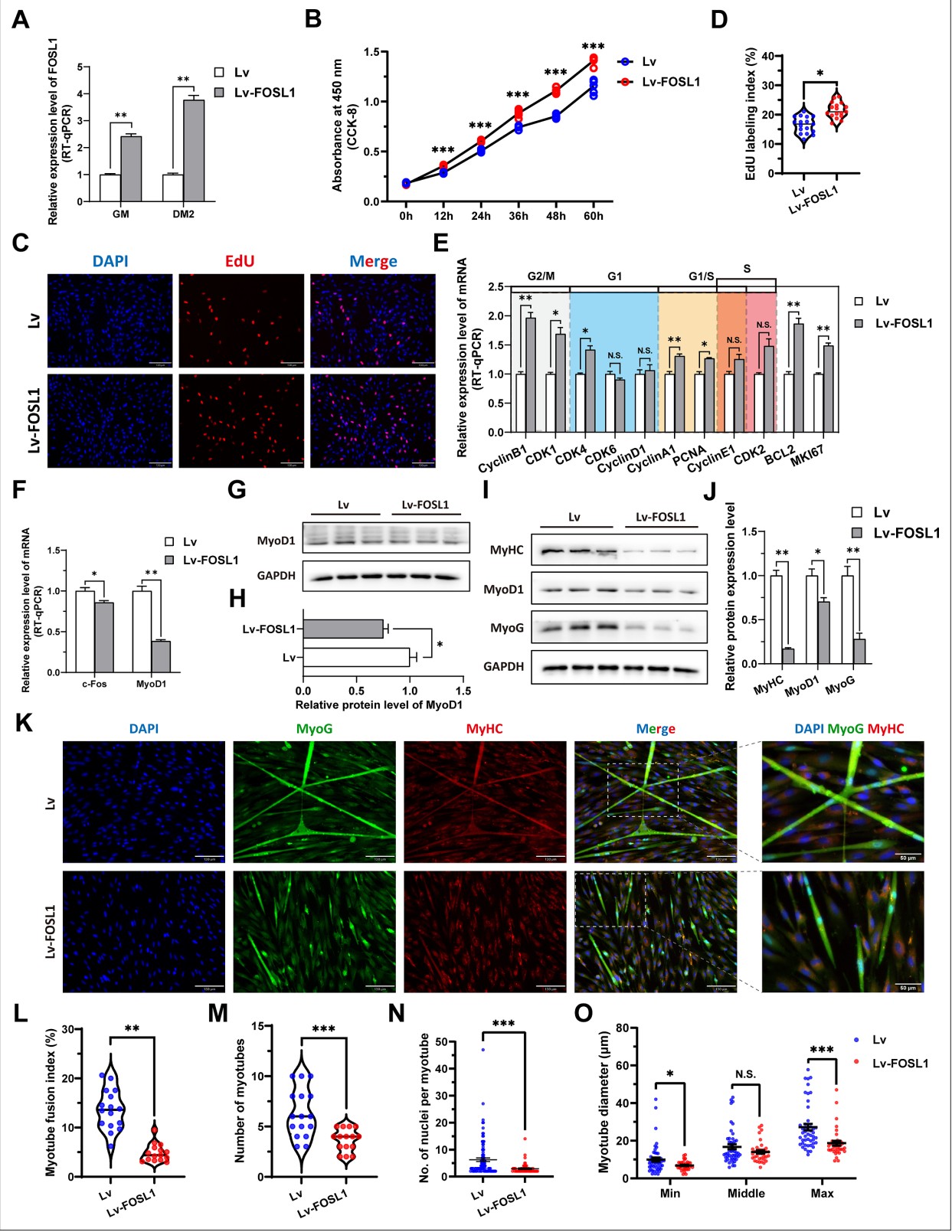

**Figure 5.** The overexpression of *FOSL1* promotes proliferation and inhibits differentiation of skeletal muscle satellite cells. (**A**) The mRNA expression level of FOSL1 at GM and DM2 after lentivirus infection (n=3). (**B**) The number of cells detected by CCK-8 at 0 hr, 12 hr, 24 hr, 36 hr, 48 hr, and 60 hr after infection with lentivirus (n=4–6). (**C–D**) EdU assay showed that the number of EdU-positive cells and EdU labeling index were significantly increased after infection with lentivirus (n=3). Scale bar 130 µm. All data points were shown. (**E**) The mRNA expression levels of cell cycle marker genes and cell

*Figure 5 continued on next page*

*Figure 5 continued*

proliferation marker genes (n=3). (**F**) The mRNA expression levels of c-Fos and MyoD1 at GM after overexpression of FOSL1 (n=3). (**G–H**) The protein expression levels of MyoD1 at GM after overexpression of FOSL1 (n=3). (**I–J**) The protein expression levels of myogenic differentiation marker genes MyoD1, MyoG and MyHC at DM2 after overexpression of FOSL1 (n=3). (**K**) The MyoG and MyHC immunofluorescence staining of myotubes at DM2 after overexpression of FOSL1. Scale bar 130 µm. (**L–O**) The myotube fusion index, number of myotubes, number of nuclei per myotube and the myotube diameter at DM2 after overexpression of FOSL1 (n=3). All data points were shown. Data: mean ± SEM. Unpaired student's t-test and chi square test were used for statistical analysis. All student's t-test were performed after the equal variance test, otherwise the t-test with Welch's correction were used. *p<0.05, **p<0.01, and ***p<0.001.

The online version of this article includes the following source data for figure 5:

**Source data 1.** Uncropped and labeled blots for *Figure 5G* and *Figure 5I*.

**Source data 2.** Raw unedited blots for *Figure 5G* and *Figure 5I*.

protein level was significantly (p<0.05) reduced in MF$^{+/-}$ cells compared to WT cells (*Figure 8A–B*), and the intracellular Ca$^{2+}$ concentration was also significantly (p<0.05) reduced (*Figure 8C–D*). The high-throughput Ca$^{2+}$ channel RYR is responsible for rapid and massive release Ca$^{2+}$ from the endoplasmic reticulum into cytoplasm. We observed a significant (p<0.05) decrease in RYR1 and/or RYR3 mRNA levels in MF$^{+/-}$ sheep skeletal muscle satellite cells and myotubes cells (*Figure 8E–F*). In addition, intracellular Ca$^{2+}$ concentration correlated with myoblasts fusion, whereas the mRNA levels of MYMK and MYMX, which control myoblasts fusion, were significantly (p<0.01) reduced in MF$^{+/-}$ cells (*Figure 8E–F*). These results suggest that the decrease Ca$^{2+}$ levels and inhibition of myoblasts fusion genes may be potential triggers for the decrease of myotube diameter and myofiber cross-sectional area in MF$^{+/-}$ sheep.

In a word, our results shed light that the *MSTN*$^{Del73C}$ mutation with *FGF5* knockout mediated the activation of FOSL1 via MEK-ERK-FOSL1 axis (*Figure 9*). The activated FOSL1 promotes skeletal muscle satellite cell proliferation and inhibits myogenic differentiation by inhibiting the expression of MyoD1, and resulting in fusion to form smaller myotubes (*Figure 9*). In addition, activated ERK1/2 may inhibit the secondary fusion of myotubes by Ca$^{2+}$-dependent CaMKII activation pathway, leading to myoblasts fusion to form smaller myotubes (*Figure 9*).

## Discussion

### Optimized Cas9 mRNA and sgRNA delivery ratio improves the efficiency of dual-gene biallelic homozygous mutations

The strategy for producing gene knockout animals by CRISPR/Cas9 gene editing system is usually to introduce the Cas9 mRNA and the sgRNA of the target gene into their prokaryotic embryos by microinjection. However, this 'one-step' method often results in a 'mosaic' of gene-edited offspring (*Wan et al., 2015*). Such chimeric mutants have now been reported in gene knockout mice (*Wang et al., 2013*), rats (*Bao et al., 2015*), monkeys (*Niu et al., 2014*), pigs (*Hai et al., 2014*), sheep (*Hongbing et al., 2014*), goats (*Wang et al., 2015*), rabbits (*Lv et al., 2016*), and humans (*Wang and Yang, 2019*) prepared by a 'one-step' method using the CRISPR/Cas9 system. For studies involved in genetic phenotypes, chimeric gene knockout animals require further cross-breeding to obtain animals with a complete knockout of the target gene. Once required to generate multiple gene knockout animals, this time-consuming and laborious operation will become extremely difficult. Although many studies have been devoted to eliminating this widespread chimeric mutation (*Sato et al., 2015*; *Sung et al., 2014*; *Kotani et al., 2015*; *Chen et al., 2015*; *Zhou et al., 2014*; *Tu et al., 2017*; *Wang et al., 2015*), however, these optimizations did not bring about a significant improvement in the production efficiency of biallelic knockout animals. Here, we increased the delivery ratio of Cas9 mRNA to sgRNA from 1:2 to 1:10, which improve the efficiency of the homozygous mutation of the biallelic gene. This unprecedented optimization method not only improved the overall gene knockout efficiency, but also the obtained gene-edited offspring were all dual-gene biallelic mutation. However, it is necessary to point out that although there are statistical differences, due to the limited number of sheep we actually produced and used for evaluation, the strategy to improve the efficiency of the homozygous mutation of biallelic gene by increasing the Cas9 and mRNA delivery ratio needs to be further comfirm in future studies.

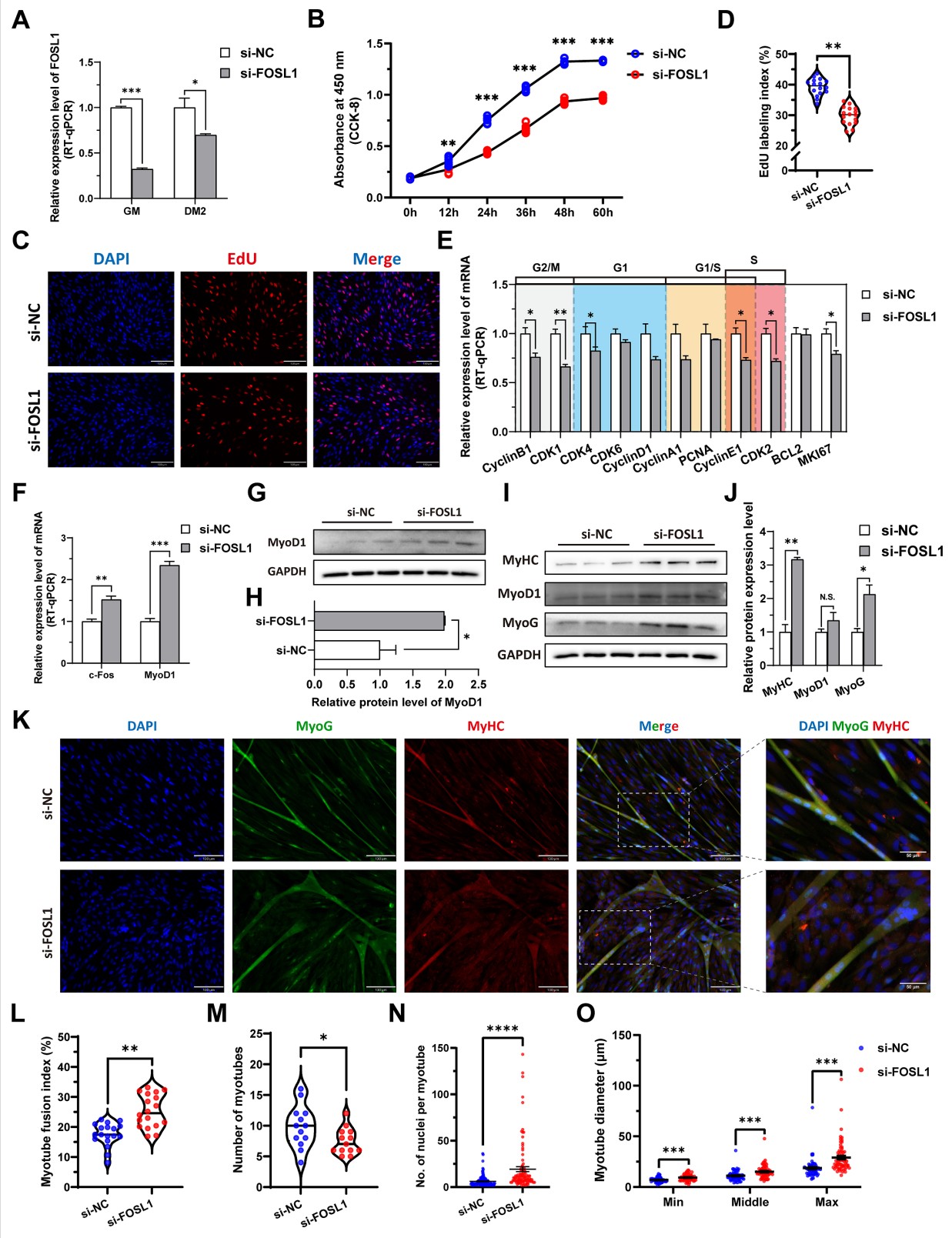

**Figure 6.** The inhibition of *FOSL1* suppresses proliferation and promotes differentiation of skeletal muscle satellite cells. (**A**) The mRNA expression level of FOSL1 at GM and DM2 after inhibiting FOSL1 (n=3). (**B**) The number of cells detected by CCK-8 at 0 hr, 12 hr, 24 hr, 36 hr, 48 hr, and 60 hr after inhibiting FOSL1 (n=4–6). (**C–D**) EdU assay showed that the number of EdU-positive cells and EdU labeling index were significantly decreased after inhibiting FOSL1 (n=3). Scale bar 130 μm. All data points were shown. (**E**) The mRNA expression levels of cell cycle marker genes and cell proliferation

*Figure 6 continued on next page*

*Figure 6 continued*

marker genes (n=3). (**F**) The mRNA expression levels of c-Fos and MyoD1 at GM after inhibiting FOSL1 (n=3). (**G–H**) The protein expression levels of MyoD1 at GM after inhibiting FOSL1 (n=3). (**I–J**) The protein expression levels of myogenic differentiation marker genes MyoD1, MyoG and MyHC at DM2 after inhibiting FOSL1 (n=3). (**K**) The MyoG and MyHC immunofluorescence staining of myotubes at DM2 after inhibiting FOSL1. Scale bar 130 μm. (**L–O**) The myotube fusion index, number of myotubes, number of nuclei per myotube and the myotube diameter at DM2 after overexpression of FOSL1 (n=3). All data points were shown. Data: mean ± SEM. Unpaired student's t-test and chi square test were used for statistical analysis. All student's t-test were performed after the equal variance test, otherwise the t-test with Welch's correction were used. *p<0.05, **p<0.01, ***p<0.001, and ****p<0.0001.

The online version of this article includes the following source data for figure 6:

**Source data 1.** Uncropped and labeled blots for *Figure 6G* and *Figure 6I*.

**Source data 2.** Raw unedited blots for *Figure 6G* and *Figure 6I*.

## Phenotypes produced by *MSTN* mutations are mutation site-dependent

As mentioned previously, although *MSTN* mutations have been found to produce a 'double-muscle' phenotype in multiple species, the microscopic phenotypes are different, and this difference is closely related to the mutation site and species types. In mice, the number of skeletal muscle fibers with *MSTN* gene knockout significantly increased by 86% (*McPherron et al., 1997*). A missense mutant *MSTN* only increased the number of mouse muscle fibers, while dominant negative *MSTN* resulted in increased muscle fiber cross-sectional area in mice, but not the number of muscle fibers (*Nishi et al., 2002*; *Zhu et al., 2000*). In addition, the use of *MSTN* neutralizing antibody on adult rats also resulted in an increased muscle fiber cross-sectional area (*Haidet et al., 2008*). In cattle, natural MSTN mutant Belgian Blue cattle had an increased number of muscle fibers and reduced muscle fiber diameter (*Wegner et al., 2000*). The muscle fiber cross-sectional area of longissimus dorsi and gluteus medius in sheep was significantly increased after a 4 bp deletion of the first exon of *MSTN* (*Zhiliang et al., 2004*). In pigs, both the *MSTN* gene-edited Meishan and Hubei pigs showed a phenotype with increased muscle fiber density (*Qian et al., 2015*; *Xu et al., 2013*). Here, we prepared *MSTN*[Del73C] mutation with *FGF5* knockout sheep with 3-base pairs of AGC in the third exon of *MSTN*, which caused the deletion of cysteine at amino acid position 73 of the mature peptide. Its macroscopic phenotype is similar to that of the MSTN-edited sheep with the first exon knocked out 4-base pairs. Both of them showed an abnormally developed 'double-muscle' phenotype of hip muscle, but the *MSTN*[Del73C] mutation with *FGF5* knockout sheep highlights a muscle fiber hyperplasia phenotype.

## FOSL1 recognizes and binds to the MyoD1 promoter and inhibits its expression

As previously described, AP-1 family members play key roles in skeletal muscle cell proliferation, differentiation, and muscle development. In this study, AP-1 family member *FOSL1* was significantly reduced in MF[+/-] sheep, and its expression were drastically reduced during myogenic differentiation, which was consistent with the decrease of *FOSL1* expression during C2C12 differentiation (*Tobin et al., 2016*). In addition, FOSL2, another AP-1 family member, can also inhibit myoblast differentiation (*Alli et al., 2013*), which may support the inhibitory effect of FOSL1 on myogenic differentiation. Therefore, *FOSL1* was recognized as a potential gatekeeper. It has been shown that *FOSL1* heterodimerizes with other transcription factors, such as the members of the bZIP family, and these dimers are either disabling the transcriptional activator complex or saving the interacting proteins from degradation in proteasomes (*Sobolev et al., 2022*). Moreover, c-Fos, a member of the AP-1 family, has been shown to inhibit MyoD1 expression and myogenesis by directly binding to the MyoD1 promoter region (*Li et al., 1992*). Therefore, we speculate that *FOSL1* may have similar functions to c-Fos. PPI analysis suggested a potential interaction between *FOSL1* and MyoD1. Subsequently, we confirmed that FOSL1 directly binds to two bZIP recognition sites in the MyoD1 promoter region, and inhibits MyoD1 promoter activity. Meanwhile, the overexpression and interference of FOSL1 further confirmed the inhibitory effect of FOSL1 on the expression of MyoD1. In a word, these results fully support our hypothesis that FOSL1 binds the MyoD1 promoter and inhibits its expression.

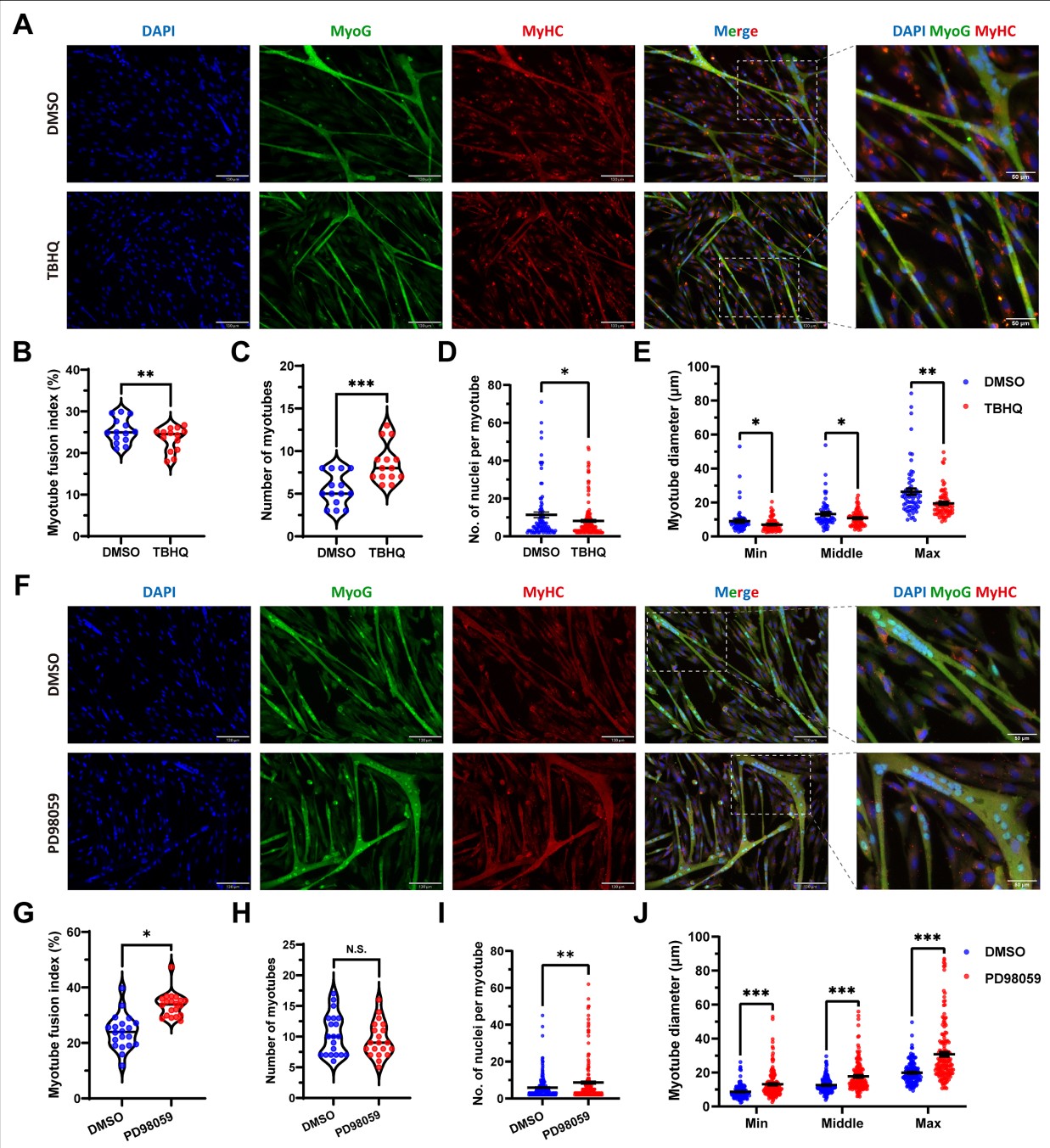

**Figure 7.** FOSL1 activity is a key regulator of myogenic differentiation and muscle myofiber hyperplasia. (**A**) Immunofluorescence staining of myogenic differentiation markers MyoG and MyHC in sheep skeletal muscle satellite cells at DM2 after addition of 20 μM TBHQ. Scale bar 130 μm or 50 μm. (**B–E**) The myotube fusion index, number of myotubes, number of nuclei per myotube and the myotube diameter at DM2 after addition of 20 μM TBHQ (n=3). All data points were shown. (**F**) Immunofluorescence staining of myogenic differentiation markers MyoG and MyHC in sheep skeletal muscle satellite cells at DM2 after addition of 1 μM PD98059. Scale bar 130 μm or 50 μm. (**G–J**) The myotube fusion index, number of myotubes, number of nuclei per myotube and the myotube diameter at DM2 after addition of 1 μM PD98059 (n=3). All data points were shown. Data: mean ± SEM. Unpaired student's t-test and chi square test were used for statistical analysis. All student's t-test were performed after the equal variance test, otherwise the t-test with Welch's correction were used. *p<0.05, **p<0.01, and ***p<0.001.

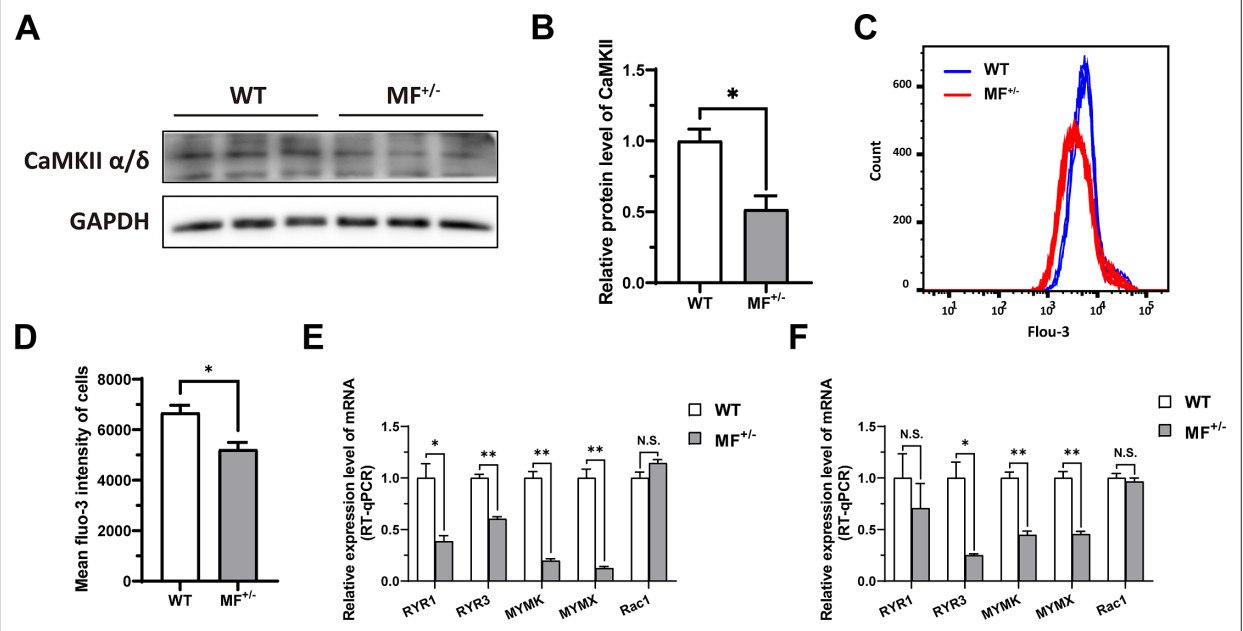

**Figure 8.** The *MSTN*^Del73C mutation with *FGF5* knockout inhibit calcium-dependent transcription signal pathway. (**A–B**) The protein expression level of CaMKII α/δ between WT and MF^+/- cells at GM (n=3). (**C**) Distribution of intracellular $Ca^{2+}$ signals between WT and MF^+/- cells at GM. (**D**) Average intracellular $Ca^{2+}$ fluorescence intensity between WT and MF^+/- cells at GM (n=4). (**E**) The mRNA expression levels of $Ca^{2+}$ channels and myoblast fusion-related genes at GM (n=3). (**F**) The mRNA expression levels of $Ca^{2+}$ channels and myoblast fusion-related genes at DM2 (n=3). Data: mean ± SEM. Unpaired student's t-test was used for statistical analysis. All student's t-test were performed after the equal variance test, otherwise the t-test with Welch's correction were used. *$p<0.05$, **$p<0.01$.

The online version of this article includes the following source data for figure 8:

**Source data 1.** Uncropped and labeled blots for *Figure 8A*.

**Source data 2.** Raw unedited blots for *Figure 8A*.

## The *MSTN*^Del73C mutation with *FGF5* knockout contribute to muscle phenotype via MEK-ERK-FOSL1 axis

The nonclassical pathway of MSTN involves PI3K/Akt/mTOR signaling pathway and MAPK signaling pathway, which mainly includes ERKs, JNKs and p38 MAPK (*Huang et al., 2007*; *Gui et al., 2012*). All of those pathways are involved in the signal transduction pathway of MSTN and mediate the transcription of MRFs (Myogenin, Myf5, MyoD), MuRF-1 and Atrogin-1, to regulate myogenic differentiation and skeletal muscle quality (*Chen et al., 2021b*). *MSTN* induces muscle fiber hypertrophy prior to satellite cell activation (*Wang and McPherron, 2012*) and inhibits IGF-I-induced increase in myotube diameter through Akt signaling pathway (*Morissette et al., 2009*). In our study, the MSTN-^Del73C mutation with FGF5 knockout resulted in the inhibition of myogenic differentiation of skeletal muscle satellite cells, and the number of myotubes and the myotube size were significantly reduced.

As previously described, the DEGs of gluteus medius RNA-seq were significantly enriched in the MAPK signaling pathway. In fact, the MAPK signaling pathway has been proven to be closely related to muscle development and myoblast differentiation (*Xie et al., 2018*; *Segalés et al., 2016*). For example, ERK1/2 promotes myoblast proliferation in response to various growth factors (*Campbell et al., 1995*), inhibits signaling pathways that activate ERK1/2, or isolates ERK1/2 in the cytoplasm, leading to cell cycle exit and cell differentiation (*Jones et al., 2001*; *Michailovici et al., 2014*). The RXR activity in myoblasts promotes myogenesis by regulating MyoD expression and acting as a MyoG cofactor (*Zhu et al., 2009*). Inhibition of MEK1/2 activates satellite cell differentiation in primary muscle fibers, and also induces myogenic differentiation and excessive fusion (*Eigler et al., 2021*). A recent study on glioma showed that FOSL1 can be activated by the Ras-MEK1/2-ERK1/2 axis in MAPK signaling pathway (*Marques et al., 2021*). Similarly, the activated MEK1/2-ERK1/2 axis in aged skeletal muscle also activates FOSL1 and increases the abundance of FOSL1 and the trans-activation capacity of the Fos-Jun heterodimer (*Mathes et al., 2021*). In our study, the *MSTN*^Del73C mutation with

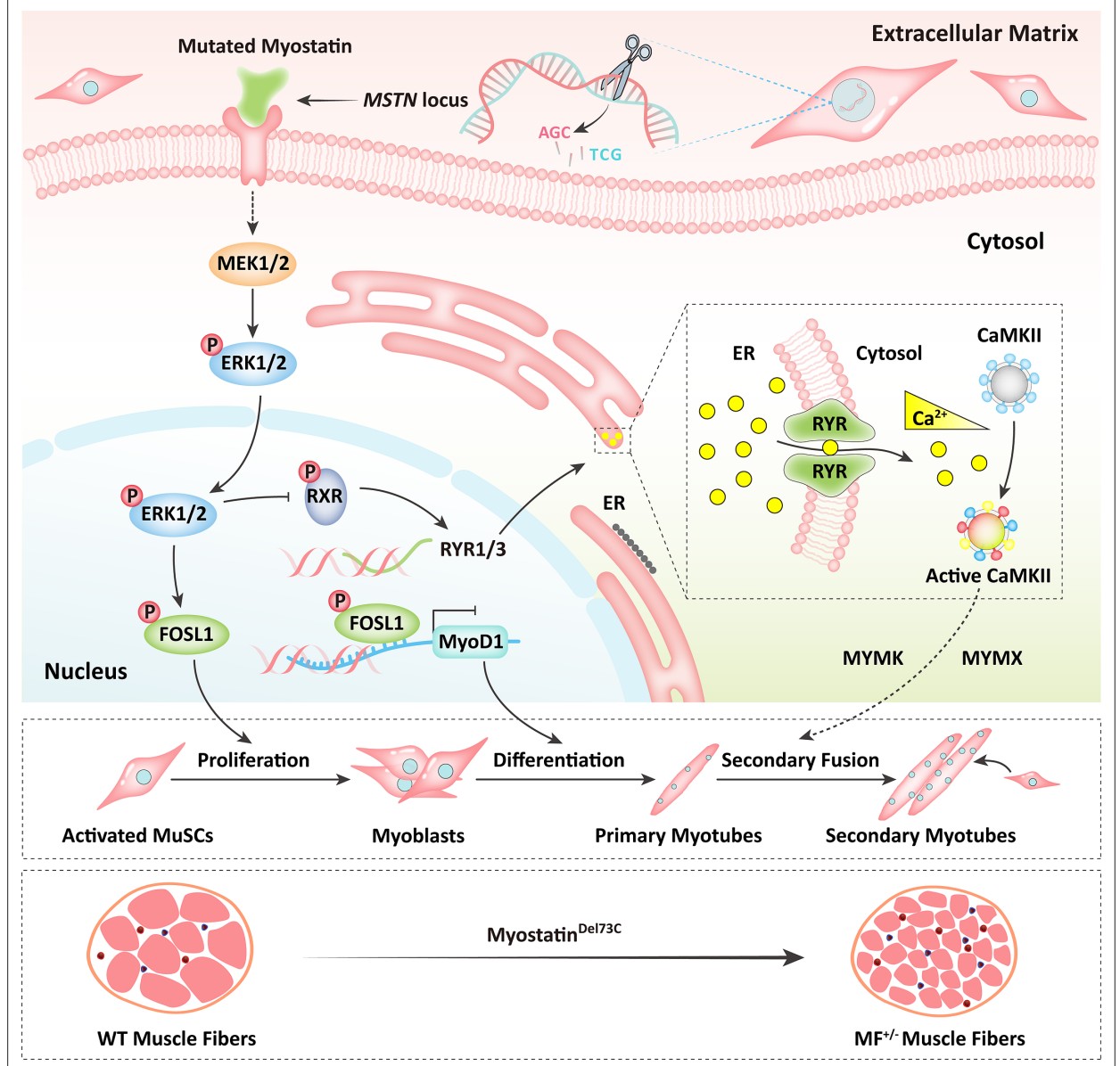

**Figure 9.** Schematic illustration of the regulation of muscle phenotypes by *MSTN*<sup>Del73C</sup> mutation with *FGF5* knockout. The *MSTN*<sup>Del73C</sup> mutation with *FGF5* knockout mediated the activation of FOSL1 via MEK-ERK-FOSL1 axis. The activated FOSL1 promotes skeletal muscle satellite cell proliferation and inhibits myogenic differentiation by inhibiting the expression of MyoD1, and resulting in fusion to form smaller myotubes. In addition, activated ERK1/2 may inhibit the secondary fusion of myotubes by Ca²⁺-dependent CaMKII activation pathway, leading to myoblasts fusion to form smaller myotubes.

*FGF5* knockout regulates FOSL1 expression and activity through MEK1/2-ERK1/2-FOSL1 axis and activated FOSL1 further inhibits myogenic differentiation of skeletal muscle satellite cells, resulting in smaller myotube diameter. However, despite the high expression of p-FOSL1 in MF⁺/⁻ myoblasts, it did not significantly inhibit the transcription of MyoD1, which may be related to a dramatic enhance in c-Fos, or there might be other parallel signaling pathways regulating MyoD1 after *MSTN*<sup>Del73C</sup> mutation with *FGF5* knockout.

Furthermore, it has been demonstrated that the inhibition of MEK1/2 using MEK1/2-specific inhibitor PD184352 can significantly down-regulate FOSL1 expression (*Mathes et al., 2021*). In our study, the indirect activation of FOSL1 by TBHQ can inhibit the myogenic differentiation of sheep skeletal muscle satellite cells, leading to reduced myoblast fusion capacity and smaller myotube diameter. In contrast, inhibition of the MEK1/2 pathway by PD98059 to suppress FOSL1 activity produced the

opposite effect. Taken together, these results shed light on the potential mechanisms by which *MSTN*-$^{Del73C}$ mutation with *FGF5* knockout leads to increased myofiber numbers and decreased fiber cross-sectional area.

## The *MSTN*$^{Del73C}$ mutation with *FGF5* knockout may affect myoblasts fusion through ERK-mediated calcium dependent transcriptional signaling pathway

$Ca^{2+}$ is recognized as a regulator of mammalian muscle fusion (*Eigler et al., 2021*; *Constantin et al., 1996*). The transient depletion of $Ca^{2+}$ in the endoplasmic reticulum is associated with myoblast differentiation and fusion (*Jones et al., 2001*). However, the signaling cascade leading to $Ca^{2+}$-mediated myoblast fusion remains unclear. Intracellular $Ca^{2+}$ level is regulated by various $Ca^{2+}$ voltage-gated channels, including but not limited to ryanodine receptors (RYR), which is the main $Ca^{2+}$ release channel of the sarcoplasmic reticulum. CaMKII is a member of the calcium/calmodulin-dependent serine/threonine kinase family. CaMKII δ, CaMKII γ, and CaMKII β are the main isoforms expressed in skeletal muscle (*Bayer et al., 1996*). Recent studies have shown that elevated intracellular $Ca^{2+}$ level is crucial for myoblasts fusion and that $Ca^{2+}$ signaling in newly formed myotubes occurs prior to the rapid growth stage of myotubes, indicating that $Ca^{2+}$ released from the endoplasmic reticulum in early myotubes may promote secondary fusion of myoblasts and myotube expansion (*Eigler et al., 2021*). Further studies showed that $Ca^{2+}$-dependent activation of CaMKII is essential for myotubes expansion and may mediate myoblast-myotube fusion by regulating MYMK and Rac1, but it is not necessary for myoblast-myoblast fusion (*Eigler et al., 2021*).

In our study, the *MSTN*$^{Del73C}$ mutation with *FGF5* knockout resulted in a significant reduction of $Ca^{2+}$ level and CaMKII α/δ protein level, and led to a decrease in myotube fusion capacity. Therefore, our results support that the increased p-ERK1/2 level promotes cell proliferation and inhibits myogenic differentiation in MF$^{+/-}$ sheep skeletal muscle satellite cells. Meanwhile, activated ERK1/2 further inhibited RYR activity by suppressing the phosphorylation of RXR, thereby reducing the release of endoplasmic reticulum $Ca^{2+}$ and potentially inhibiting the secondary fusion of myotubes by $Ca^{2+}$-dependent CaMKII activation pathway, and further mediating myofiber hyperplasia.

## Methods

### Production of Cas9 mRNA and sgRNA

The Cas9 and U6-sgRNA co-expression vector backbones pX330 were purchased from Addgene (plasmid ID: 42230 and 48138). Sheep MSTN and FGF5 sgRNAs were designed using CRISPR Design Tool (http://tools.genome-engineering.org). The MSTN sgRNA (GACATCTTTGTAGGAGTACAGCAA) and FGF5 sgRNA (AGGTTCCCCTTTCCGCACCT) were used in this study. Two complementary guide sequence oligos were synthesized, annealed, and cloned into the pX330 backbone vector to form the functional co-expression plasmids. T7 promoter was linked to the 5' ends of the Cas9 coding region and MSTN/FGF5 sgRNA template by PCR amplification from the pX330-MSTN/FGF5 plasmid constructed as described above. Then, these PCR products, as transcription templates, were purified using E.Z.N.A. Cycle Pure Kit (Omega Bio-Tek). MSTN sgRNA and FGF5 sgRNA were prepared by in vitro transcription (IVT) using the MEGAshortscript T7 Kit (Life Technologies). Cas9 mRNA was transcribed with the m7G(5')ppp(5') G cap on its 5' terminal and poly (A) tail on its 3' terminal using the mMESSAGE mMACHINE T7 Ultra Kit (Life Technologies). Both the Cas9 mRNA and the sgRNAs were purified using the MEGAclear Kit (Life Technologies) and eluted in RNase-free water.

### Microinjection and embryo transfer

The procedure for the efficient production of pronuclear embryos has been described previously (*Li et al., 2016*). Briefly, Cas9 mRNA (1000 ng/μL) and sgRNAs (200 ng/μL) were mixed and injected into the transferable embryos in which the zona pellucida was clear, cytoplasm was uniform, and pronucleus was visible using a FemtoJet microinjector (Eppendorf). Following microinjection, three to five embryos were transplanted into the oviduct of each recipient within 1 hr after starting the laparotomy operation. Pregnancy was confirmed by transabdominal ultrasound scanning on the 60th day after embryo transfer.

## Tissue sample collection and preparation

Gluteus medius and longissimus dorsi were harvested from WT and *MSTN*[Del73C] mutation with *FGF5* knockout (MF[-/-]) sheep, and three WT sheep and four MF[-/-] F1 generation sheep (half-sib) were used for feeding and slaughter. All sheep were female and are slaughtered at 12-month-old. All samples were immediately frozen in liquid nitrogen and then stored at −80 °C until analysis. All sheep are raised by the national feeding standard NT/T8152004. All procedures performed for this study were consistent with the National Research Council Guide for the Care and Use of Laboratory Animals. All experimental animal protocols in this study were approved and performed following the requirements of the Animal Care and Use Committee at China Agricultural University (AW02012202-1-3). All surgeries were performed under sodium pentobarbital anesthesia, and all efforts were made to minimize any suffering experienced by the animals used in this study.

## H&E staining and morphological analysis of muscle fibers

Fresh muscle tissue samples were fixed, dehydrated, embedded and frozen sectioned, respectively. Next, they were sequentially stained with hematoxylin and eosin, then dehydrated with gradient ethanol and transparentized with xylene, and finally sealed with neutral resin to make tissue sections. The images from at least five random fields were captured with an inverted microscope. The Image J software was used to segment, count and calculate the area of each muscle fiber cell. The number of fiber cell in a fixed area size was calculated to estimate the number of fiber cell per unit area.

## Cell isolation, culture, and transfection

Sheep skeletal muscle satellite cells were isolated and cultured as previously described (*Chen et al., 2021a*). In brief, the muscle tissues of the hind limbs from 3-month-old sheep fetuses were cut into small pieces, digested sequentially with 0.2% collagenase type II (Gibco, Grand Island, NY) and 0.25% trypsin (Gbico, Grand Island, NY). The cell suspension was successively filtered through 100, 200, and 400 mesh cell sieves. After this, the cells were resuspended in growth medium (GM) containing DMEM/F12 (Gbico, Grand Island, NY) with 20% fetal bovine serum (FBS, Gibco) and 1% penicillin-streptomycin liquid (Gbico, Grand Island, NY), and cultured for 2–3 times with differential adhesion. To induce differentiation, the cells were cultured to 70% confluence in GM, and followed by an exchange to differentiation medium (DM) containing DMEM high glucose (Gbico, Grand Island, NY) with 2% horse serum (HS, Gibco) and 1% penicillin-streptomycin. To produce viral solution for over-expression of the target gene, it was subcloned into the XbaI and BamHI sites of the lentiviral vector by seamless cloning, and the primer sequences of gene cloning were listed in *Supplementary file 2A*. HEK 293T cells were co-transfected with the envelope plasmid pMD2.G, the packaging plasmid psPAX2 and the target plasmid at a mass ratio of 1:2:4 to produce the virus. The siRNA were synthesized by Guangzhou RiboBio Co., Ltd, and the sequences were listed in *Supplementary file 2B*. Then, the cells were infected with packaged lentivirus or transfected with siRNA using Lipofectamine 3000 (Invitrogen, USA) when they were cultured to 60–70% confluence.

## Total RNA isolation and real-time quantitative PCR (RT- qPCR)

The total RNA of tissues and cells was isolated using TRIzol reagent (Sangon Biotech, Shanghai, China) following the manufacturer's protocol. In short, after tissues or cells were lysed, chloroform was added to separate the organic and inorganic phases, followed by precipitation with isopropanol and ethanol in turn, and finally, the RNA was dissolved in DEPC water. Then, the first strand cDNA was prepared using PrimeScript II 1st Strand cDNA Synthesis Kit (Takara, Beijing, China). qPCR was performed using 2×SYBR Green qPCR Mix (Low ROX) (Aidlab Biotechnologies, Beijing, Chian) in a Stratagene Mx3000P (Agilent Technologies, SUA). With GAPDH mRNA as endogenous control, the relative expression level of genes was calculated by the $2^{-\Delta\Delta Ct}$ method. All primers used were listed in *Supplementary file 2C*.

## Western blot

Tissue or cell samples were lysed in RIPA buffer (Solarbio, Beijing, China) supplemented with protease and phosphatase inhibitor cocktail (Beyotime, Beijing, China) for total protein extraction. Then, equal amounts of tissue or cell lysate were resolved by 10% SDS-PAGE and transferred onto PVDF membranes (Millipore, USA). The membranes were blocked with 5% BSA for 1 hr, incubated with primary antibody at 4 °C overnight, then incubated with secondary antibody for 1 hr before detection.

The fold change of protein was normalized to GAPDH for quantitative analysis by ImageJ software. The antibodies information was listed in *Supplementary file 2D*.

## 5-Ethynyl-2′-deoxyuridine (EdU) assay

At 24 hr after transfection, sheep skeletal muscle satellite cells were incubated at 37 °C for 2 hr in 96-well plates with 50 μM EdU (RiboBio, Guangzhou, China). Then, fixed the cells in 4% paraformaldehyde for 30 min and neutralized using 2 mg/mL glycine solution. The Apollo staining solution which contains EdU was added and incubated at room temperature for 30 min in the dark to label the DNA in the synthesis stage, the nuclear was then counterstained with DAPI. The number of EdU positive cells was counted from the images of five random fields obtained with an inverted fluorescence microscope at a magnification of 100×. EdU labeling index was expressed as the number of EdU-positive cell nuclei/total cell nuclei.

## Cell counting kit-8 (CCK-8) and cell cycle detection

Skeletal muscle satellite cells were seeded in 96-well plates and cultured for appropriate time according to different experimental treatments. Then, 10 μL CCK-8 solution was added to each well and incubated at 37 °C in a 5% $CO_2$ incubator for 2 hr, and then the absorbance at 450 nm was measured with a microplate reader. The cultured skeletal muscle satellite cells were digested with trypsin, centrifuged at $1000 \times g$ for 5 min to collect the cell pellet, washed once with ice-cold PBS, and then 1 mL of ice-cold 70% ethanol was added to fix the cells overnight at 4 °C. The next day, the cells were washed with ice-cold PBS again, and the cells were incubated with 0.5 mL PI staining solution at 37 °C for 30 min and collected by flow cytometry at low speed.

## Immunofluorescence staining

Sheep skeletal muscle cells were fixed in 4% paraformaldehyde for 30 min, permeabilized in 0.1% Triton X-100 for 20 min and blocked with 5% normal goat serum for 30 min at room temperature, and then incubated with primary antibody at 4°C overnight. Next, the fluorescent secondary antibody was added and incubated at 37°C for 1 hr in the dark, and the nuclear was then counterstained with DAPI. The immunofluorescence images from five random fields were captured with an inverted fluorescence microscope.

## Chromatin Immunoprecipitation (ChIP)

The cells were fixed with 1% formaldehyde, then the cells were collected with a cell scraper and resuspended in cell lysis buffer (10 mM HEPES, 0.5% NP-40, 1.5 mM MgCl2, 10 mM KCl, pH 7.9) containing protease inhibitor cocktail (Beyotime, Beijing, China) and incubated on ice to release the cytoplasm. Next, cell pellets were collected and resuspended in nuclear lysis buffer (50 mM Tris, 10 mM EDTA, 0.3% SDS, pH 8.0) containing protease inhibitor cocktail. After the DNA was fragmented by ultrasonication, the supernatant was collected. The samples were diluted ChIP dilution buffer (0.01% SDS, 1.1% Triton X- 100, 1.2 mM EDTA, 16.7 mM Tris-HCl pH 8.0, 167 mM NaCl), then 5 μg primary antibody was added and incubated overnight at 4 °C with rotation. The next day, protein A/G magnetic beads were added to each sample and incubated at 4 °C with rotation for 2 hr. Then, the magnetic beads were respectively washed once with low-salt wash buffer (0.1% SDS, 1% Triton X-100, 2 mM EDTA, 20 mM Tris-HCl pH 8.0, 150 mM NaCl), high-salt wash buffer (0.1% SDS, 1% Triton X-100, 2 mM EDTA, 500 mM NaCl, 20 mM Tris-HCl pH 8.0), LiCl wash buffer (0.25 M LiCl, 1% NP-40, 1% sodium deoxycholate, 1 mM EDTA, 10 mM Tris-HCl pH 8.0), and TE buffer (10 mM Tris-HCl pH 8.0, 1 mM EDTA) at 4 °C. Next, ChIP elution buffer (1% SDS, 100 mM $NaHCO_3$) containing proteinase K was added to each sample, then incubated at 62 °C overnight, and the DNA was finally purified by a purification column.

## RNA-seq

The rRNA was removed from each total RNA sample of the gluteus medius to construct a strand-specific transcriptome sequencing library, and the Illumina Novaseq 6000 sequencing platform was used to perform high-throughput sequencing with a paired-end read length of 150 bp. Raw data were transformed into clean reads by removing reads containing adapter, ploy-N and low-quality reads from raw data. At the same time, Q20, Q30, GC-content, and sequence duplication levels of the clean

data were calculated. The genome index was constructed using Hisat2 software and the clean reads were mapped to the sheep reference genome (*Oar Rambouillet v1.0*), the featureCounts software was used for expression quantification, and DESeq2 software was used for differential expression analysis based on p-value <0.05 and | log2 Fold Change |>1.

## Conclusion

In this study, we found that increasing the delivery ratio of Cas9 mRNA to sgRNA can improve the efficiency of the homozygous mutation of the biallelic gene. Based on this, we generated a *MSTN*$^{Del73C}$ mutation with *FGF5* knockout sheep, a dual-gene biallelic homozygous mutant, which highlights a dominant 'double-muscle' phenotype. Both F0 and F1 generation mutants highlight the excellent trait of high-yield meat and the more number of muscle fibers per unit area. Our results suggested the *MSTN*$^{Del73C}$ mutation with *FGF5* knockout mediated the activation of FOSL1 via MEK-ERK-FOSL1 axis, further promotes skeletal muscle satellite cell proliferation, and inhibits myogenic differentiation by inhibiting the expression of MyoD1, and resulting in smaller myotubes. In addition, activated ERK1/2 may inhibit the secondary fusion of myotubes by $Ca^{2+}$-dependent CaMKII activation pathway, leading to myoblasts fusion to form smaller myotubes. This supports the myofiber hyperplasia that more number of muscle fibers and smaller cross sectional area, caused by the MSTN$^{Del73C}$ mutation with FGF5 knockout.

## Acknowledgements

This work was supported by Major Agricultural Biological Breeding Project (2022ZD04014), National Natural Science Foundation of China (32072722, 32272853), and National Key Research and Development Program-Key Projects (2021YFF1000704, 2021YFD1200902).

## Additional information

### Funding

| Funder | Grant reference number | Author |
|---|---|---|
| Major Agricultural Biological Breeding Project | 2022ZD04014 | Zheng-Xing Lian |
| National Natural Science Foundation of China | 32072722 | Kun Yu |
| National Natural Science Foundation of China | 32272853 | Zheng-Xing Lian |
| National Key Research and Development Program of China | 2021YFF1000704 | Kun Yu |
| National Key Research and Development Program of China | 2021YFD1200902 | Zheng-Xing Lian |

The funders had no role in study design, data collection and interpretation, or the decision to submit the work for publication.

### Author contributions

Ming-Ming Chen, Data curation, Visualization, Writing – original draft, Writing – review and editing; Yue Zhao, Data curation, Formal analysis, Investigation, Validation, Writing – review and editing; Kun Yu, Methodology, Project administration, Writing – review and editing; Xue-Ling Xu, Data curation, Formal analysis, Validation, Writing – review and editing; Xiao-Sheng Zhang, Jin-Long Zhang, Xiao-Fei Guo, Guo-Shi Liu, Methodology, Resources; Su-Jun Wu, Formal analysis, Investigation; Zhi-Mei Liu, Data curation, Validation; Yi-Ming Yuan, Formal analysis, Software; Shi-Yu Qi, Data curation, Investigation; Guang Yi, Data curation, Formal analysis; Shu-Qi Wang, Validation; Huang-Xiang Li, Ao-Wu Wu, Investigation; Shou-Long Deng, Conceptualization, Formal analysis, Writing – review and editing; Hong-Bing Han, Conceptualization, Methodology; Feng-Hua Lv, Methodology, Supervision, Writing

– review and editing; Di Lian, Data curation, Formal analysis, Methodology, Writing – review and editing; Zheng-Xing Lian, Funding acquisition, Project administration, Supervision, Writing – review and editing

**Author ORCIDs**
Ming-Ming Chen ⬛ http://orcid.org/0000-0003-2059-6342
Zheng-Xing Lian ⬛ http://orcid.org/0000-0001-8418-4916

Reviewer #3 (Public review): https://doi.org/10.7554/eLife.86827.3.sa1
Author response https://doi.org/10.7554/eLife.86827.3.sa2

## Additional files

**Supplementary files**
• Supplementary file 1. Production and slaughter information of $MSTN^{Del73C}$ mutation with $FGF5$ knockout sheep. (A) Summary of generation of sheep carrying biallelic mutations in dual genes via the CRISPR/Cas9 system. (B) Muscle weight of different parts in WT and MF$^{+/-}$ sheep (g). (C) The slaughter traits of muscles in WT and MF$^{+/-}$ sheep. (D) Meat quality of longissimus dorsi in WT and MF$^{+/-}$ sheep. (E) Shearing force of different parts in WT and MF$^{+/-}$ sheep (N). (F) Amino acid content of longissimus dorsi in WT and MF$^{+/-}$ sheep (%, DM basis).

• Supplementary file 2. The information of primers sequence, siRNA sequence and antibody. (A) Primers sequences of gene cloning. (B) The sequences of siRNA. (C) All primers of PCR and RT-qPCR. (D) The antibodies information.

• MDAR checklist

### Data availability

The raw sequence data reported in this paper have been deposited in the Genome Sequence Archive (*Chen et al., 2021c*) in National Genomics Data Center (*CNCB-NGDC Members and Partners, 2022*), China National Center for Bioinformation / Beijing Institute of Genomics, Chinese Academy of Sciences (GSA: CRA008539) that are publicly accessible at https://ngdc.cncb.ac.cn/gsa.

The following dataset was generated:

| Author(s) | Year | Dataset title | Dataset URL | Database and Identifier |
|---|---|---|---|---|
| Chen MM | 2022 | Sheep skeletal muscle RNA-seq | https://ngdc.cncb.ac.cn/gsa/search?searchTerm=CRA008539 | Genome Sequence Archive, CRA008539 |

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
