## [Editor Report · eLife assessment]

The authors present a **useful** analysis of the phenotype of sheep in which the muscle developmental regulator myostatin has been mutated in a FGF5 knockout background. The goal was to produce sheep with a "double-muscled" phenotype, yet the genetically engineered sheep exhibited meat with a smaller cross-sectional area and higher number of muscle fibers. The work extends the extensive body of knowledge already published in this area. The authors provide evidence using in vitro experiments that Fosl1 regulates myogenesis, but the strength of evidence relating to the muscle phenotype and underlying cellular and molecular mechanism remains **incomplete**.

---

## [Referee Report · Reviewer #3 (Public review)]

Although the authors findings are interesting, they do little to demonstrate new scientific information or advancements in producing genetically modified livestock with improved production characteristics. While the MSTNDel273 sheep exhibited an increased number of muscle fibers, the data provided did not demonstrate a significant improvement in meat production, quality or quantity in the MSTNDel273 sheep vs WT.

The manuscript is very long, complicated and difficult to read, given the minimum amount of significant information that is provided. It reads more like a graduate student thesis than a scientific manuscript ready for publication. Given the significant findings are so minimal, the amount of text provided, figures and tables are excessive. A large number of different molecular techniques are employed to try and decipher the mechanism(s) that result in the observed phenotype = double muscling. The authors focus on the MEK-ERK-FOSL1 pathway and suggest this is the key pathway/mechanism resulting in the phenotype observed in MSTNDel273sheep. However, they provide very little "significant" evidence to support this. RNA-Seq data demonstrated that hundreds of different genes were either upregulated or down-regulated, but the authors chose to only focus on FOSL1 and associated genes. The findings do not support the idea that FOSL1 is not involved, but neither do they strongly support FOSL1 involvement. The observations made by the authors could be co-incidental and not causative in nature.

The authors indicate that sgRNA design changes in addition to changing the molar ratio of Cas9MRNA:sgRNA improved the ability to generate biallelic homozygous mutant sheep; however, the data provided to not demonstrate any significant difference. Given the small number of sheep that were actually produced and evaluated, it is extremely difficult to demonstrate anything that was analyzed to be significantly (statistically) different between MSTNDel273 sheep and WT, yet the authors seem to ignore this in much of their discussion. There is no explanation as to why the authors started with sheep that were FGF5 knockouts. The reviewer assumes that this was simply a line of sheep available from previous studies and the goal was to produce sheep with both improved hair/wool characteristics in addition to improved muscle development. However, the use of FGF5 knockout sheep complicates the ability to accurately decipher the unique aspects associated with targeting only myostatin for knock-out. At minimum, this is a variable that has to be considered in the statistical analysis. No information is provided on the methods used to produce the MSTNDel273 sheep, which seems fundamentally important. It is assumed they were produced by injecting one-cell zygotes then transferring these into surrogate females, but given the information provided, it is impossible to know. Certainly, the methods employed could have a profound effect on the outcome. There is no information provided on the sex of the animals produced and then analyzed.

Comments on revised version:

The manuscript by Chen et al. is improved and demonstrates successful gene editing in sheep embryos to obtain biallelic mutation of Mstn and FGF5. Despite the improvements in the revised manuscript, the cellular and molecular mechanism remain inadequate to conclude whether Fosl1 indeed acts downstream of myostatin. In addition, there is little that is new direction versus confirmatory for what is already well know regarding Mstn and FGF5

There are also a number of editorial mistakes e.g. the authors refer to tables S1-S4 in the materials and methods and results section, but there is no table S1-S4 provided.

---

## [Author Response]

The following is the authors’ response to the original reviews.

**eLife assessment**
The authors present a useful analysis of the phenotype of sheep in which the muscle developmental regulator myostatin has been mutated in a FGF5 knockout background. The goal was to produce sheep with a "double-muscled" phenotype, yet the genetically engineered sheep exhibited meat with a smaller cross-sectional area and higher number of muscle fibers. The work extends the extensive body of knowledge already published in this area. The authors provide evidence using in vitro experiments that Fosl1 regulates myogenesis, but the strength of evidence relating to the muscle phenotype and underlying cellular and molecular mechanism is inadequate.

Thanks for assessment. According to the reviewers' comments, we have supplemented and updated the data on muscle phenotypes, and the molecular mechanisms also have been supplemented accordingly, such as FOSL1 silencing and inhibition, as as well as possible secondary fusion of myoblasts regulated by calcium signaling. Meanwhile, considering the suggestions of editors and reviewers, we have also supplemented the data on serum MSTN regulation. Given that the phenotype of MSTN gene editing is mutation site dependent, we directly cultured skeletal muscle satellite cells using serum from WT and MF^+/-^ sheep, and showed that the serum regulation cannot be ignored after *MSTN*^Del73C^ mutation with *FGF5* knockout.

**Public Review:**
Chen and collaborators first analysed in sheep embryonic gene editing using CRISPR-Cas9 technology to invalidate the two alleles of Mstn and Fgf5 genes by using different ratios of Cas9 mRNA and sgRNA. They showed that a ratio of 1:10 had highest efficiency and they successfully generated two sheep with biallelic mutations of both genes. Materials and Methods on the generation of gened edited sheep is entirely missing. The data on these gene edited sheep have been already published twice by the authors in different contexts. Other groups reported on gene editing of Mstn or Fgf5 in sheep embryos and the resulting phenotypes.

We thank the reviewers for pointing out our negligence and shortcomings. We have provided detailed information on the generation method of gene editing sheep in the Materials and Methods. Briefly, gene-edited sheep were produced by injecting MSTN sgRNA, FGF5 sgRNA, and Cas9 mRNA into embryos in different ratio.

Although the findings are interesting, they do not provide sufficiently new scientific information or advancements in producing genetically modified livestock with improved production characteristics. While the MSTNDel273 sheep exhibited an increased number of muscle fibers, the data provided did not demonstrate a significant improvement in meat productions, quality or quantity in the MSTNDel273 sheep vs WT.

Thank you very much for your constructive comments. Considering the lack of data on improving production traits, we have further supplemented the data on meat yield and quality of *MSTN*^Del73C^ mutation with *FGF5* knockout sheep in Supplementary file 1B-F. Although these improvements were not significant enough, our data showed increased meat production traits in *MSTN*^Del73C^ mutation with *FGF5* knockout sheep, such as the proportion of hind leg meat to carcass and the proportion of gluteus medius to carcass. For example, the proportion of hind leg meat was significantly increased by 21.2% (Supplementary file 1C), and the proportion of gluteus medius in the carcass of MF^+/-^ sheep was significantly (*P*<0.01) increased by 26.3% compared to WT sheep (Figure 2K). In addition, there were no significant (*P*>0.05) differences in pH, color, drip loss, cooking loss, shearing force, and amino acid content of the longissimus dorsi between WT and MF^+/-^ sheep (Supplementary file 1D-F). All these results demonstrated that the *MSTN*^Del73C^ mutation with *FGF5* knockout sheep had well-developed hip muscles with smaller muscle fibers, which do not affect meat quality, and this phenotype may be dominated by *MSTN* gene.

The authors indicate that sgRNA design changes in addition to changing the molar ratio of Cas9MRNA:sgRNA improved the ability to generate biallelic homozygous mutant sheep; however, the data provided to not demonstrate any significant difference. Given the small number of sheep that were actually produced and evaluated,it is extremely difficult to demonstrate anything that was analyzed to be significantly (statistically) different between MSTNDel273 sheep and WT, yet the authors seem to ignore this in much of their discussion. There is no explanation as to why the authors started with sheep that were FGF5 knockouts. The reviewer assumes that this was simply a line of sheep available from previous studies and the goal was to produce sheep with both improved hair/wool characteristics in addition to improved muscle development. However, the use of FGF5 knockout sheep complicates the ability to accurately decipher the unique aspects associated with targeting only myostatin for knock-out. At minimum, this is a variable that has to be considered in the statistical analysis. No information is provided on the methods used to produce the MSTNDel273 sheep, which is fundamentally important. It is assumed they were produced by injecting one-cell zygotes then transferring these into surrogate females. The methods employed might have a profound effect on the outcome.

We greatly appreciate your review. In the current study, we did not discuss the impact of changes in sgRNA design on the ability to generate biallelic homozygous mutant sheep. In fact, we focused on the delivery molar ratio of Cas9 mRNA to sgRNA and found that increasing the molar ratio of Cas9:sgRNA can improve the ability to produce homozygous biallelic mutations in sheep. We apologize for neglecting this statistical analysis, which was tested for significance of differences in the revised version by the chi-square test. Other restrictions related to the actual production and evaluation of the number of sheep were analyzed in our additional discussion. It should be explained to the reviewers that the gene-edited sheep we produced did not start with FGF5 knockout sheep. As hypothesized by the reviewers, we used a one-step method to simultaneously edit the two genes of MSTN and FGF5 to concomitantly increase muscle yield and improve wool characteristics in sheep, which resulted in knockout of the FGF5 gene and mutation of the MSTN gene. As speculated by the reviewers, the *MSTN*^Del73C^ mutation with *FGF5* knockout sheep was generated by injecting sgRNA and Cas9 mRNA of MSTN and FGF5 into a single fertilized egg and then transplanted into a surrogate mother. We have provided detailed information on the generation method of gene edited sheep in the Materials and Methods section.

Authors genotyped one sheep with a biallelic three base pair deletion in Mstn exon 3 and a compound heterozygote mutation in Fgf5 with a 5 nucleotides deletion on one allele and 37 nucleotides deletion on the other allele, partially spanning over the same region. This sheep developed a double muscle phenotype, which was documented using photography and CT scan. The hair phenotype was not further addressed, but authors referred to a previous publication.

Thank you for your review. In the current study, we only focused our perspective on the muscle phenotype, while the data on the hair phenotype involved another study. Therefore, we referred to our previous publication on hair phenotypes, in which the mutation locus in FGF5 gene-edited sheep is the same as in the current study.

Authors performed morphometric studies on two distinct muscles, longissimus dorsi and gluteus medius, and found a profound fiber hypotrophy in the Mstn-/-;Fgf5-/- double mutants, with a shift from larger fiber diameter to smaller fiber sizes. Morphometric studies showed only a low percentage of fibers in wt and mutant sheep had fiber cross sectional areas larger than 800 µm2, whereas about 30% in wt and about 60% in the mutant had CSA of <400 µm2. The report of one case, without reproducing the phenotype in other sheep, is scientifically insufficient. The fiber sizes in wt sheep remains far below previously published reports in sheep (about 3-5 times smaller) and as compared to other species, which suggests a methodological error in morphometric methods.

We greatly appreciate your careful review. There is indeed an error in morphological analysis of the MF^-/-^ sheep longissimus dorsi and gluteus medius muscles. After carefully checked, we found that the reason for the fiber sizes in WT sheep remains far below previously published reports in sheep was due to the incorrect use of scale. Thus, we re-scanned the tissue sections and re-calculate the cross-sectional area of muscle fibers and the number of muscle fiber cells per unit area with the correct scale. In this case, the average cross-sectional area of muscle fibers in WT sheep was approximately 1800 μm^2^, which is consistent with the previous report. We once again salute the reviewing expert for such a careful and conscientious review. Considering the profound fiber hypotrophy in *MSTN*^Del73C^ mutation with *FGF5* knockout sheep as pointed out by the reviewer, we performed a statistical analysis on the proportion of centrally nucleated myofibres between WT and MF^+/-^ sheep, which can characterize the occurrence of muscle fiber hypotrophy. The results showed that there was no significant difference in the proportion of centrally nucleated myofibres between WT and MF^+/-^ sheep (Figure 2-figure supplement 2D). At the same time, we also analyzed the mRNA expression levels of muscle fiber hypotrophy and muscle atrophy related genes, such as MTM1, DMD, IGF1, SMN1, and GAA. Although the levels of MTM1, IGF1, SMN1, and GAA were significantly increased (Figure 2-figure supplement 2E), this elevation did not result in the occurrence of muscle fiber hypotrophy and muscle atrophy, but was beneficial for muscle formation. Therefore, we suggest that the phenomenon produced by *MSTN*^Del73C^ mutation with *FGF5* knockout may not be muscle fiber hypotrophy. Because *MSTN*^Del73C^ mutation with *FGF5* knockout significantly promotes the proliferation of sheep skeletal muscle satellite cells (Figure 3A-F), and more importantly, its muscle phenotype in MF^-/-^ and MF^+/-^ sheep were improved, including the "double-muscle" phenotype of the rump (Figure 2A), the proportion of gluteus medius in the carcass (Figure 2K), and the proportion of hind leg meat (Supplementary file 1C).

The authors also investigated the influence of Fgf5 mutation on muscle development. They determined fiber cross sectional area in heterozygous Fgf5 mutant (number of investigated animals not given) and conclude that Mstn mutation but not Fgf5 mutation caused the double muscle phenotype. Results are insufficient to support this conclusion. Firstly, authors investigated heterozygous FGF5 sheep and not homozygous mutants. Secondly, FGF5 has previously been shown to stimulate expansion of connective tissue fibroblasts and to inhibit skeletal muscle development during limb embryonic development (Clase et al. 2000). Of note, Mstn is also expressed during embryonic development. A combined knockout could therefore entail synergistic effects and cause muscle hyperplasia that is not found in individual knockout, a hypothesis that was not addressed by the authors.

Thank you very much for your critical review, which is very valuable for improving the quality of our manuscript. We have given the number of animals studied in all figure legends. Given the lack of MSTN and FGF5 single gene edited sheep, both homozygous and heterozygous sheep, especially MSTN single gene edited sheep, we have weakened the view that MSTN mutations rather than FGF5 mutations lead to “double-muscle” phenotype in conclusion and discussion. As you have mentioned, our current data is indeed insufficient to support this conclusion. In addition, considering the expression of MSTN and FGF5 in embryonic development and their regulation of skeletal muscle development, we examined the expression of MSTN and FGF5 in individual development after *MSTN*^Del73C^ mutation with *FGF5* knockout (Figure 2-figure supplement 2A). However, these results are limited by the animals involved in embryonic development, especially single gene edited embryos. We greatly appreciate your very meaningful and valuable comments on the possible synergistic effects of combined knockdown. We will prepare MSTN and FGF5 single gene edited sheep to further explore possible synergistic effects in the following study.

The authors generated and studied an F1 generation of mutant sheep with heterozyogous mutation in Mstn and Fgf5. In Mstn+/-;Fgf5+/-, gluteus medius muscle was found to be larger compared to wt sheep, whereas other muscles were smaller, and overall meat quantity did not change. Morphometric studies revealed a similar muscle fiber hypotrophy and muscle hyperplasia as in the Mstn-/-;Fgf5-/- gluteus muscle.

Thank you for your comments. We found that the proportion of gluteus medius in MF^+/-^ sheep was larger than that in WT sheep, and in addition, the proportion of hind leg meat also significantly increased (Supplementary file 1C). Morphological analysis shows that MF^+/-^ sheep exhibited a myofiber hyperplasia phenotype similar to MF^-/-^ sheep.

In the next part of results, authors investigated the presence of myostatin protein in homozygous Mstn muscle using immunohistochemistry and found no differences compared to wt, however, positive and negative controls are missing. The also determined Mstn transcription and protein quantity using WB in heterozygous Mstn muscle and found no difference. The authors did not provide data to explain of why the herein generated Mstn mutation causes muscle fiber hypotrophy, whereas most work on myostatin abrogation demonstrated fiber hypertrophy.

Thank you very much for your constructive comments. Due to the lack of necessary positive and negative controls in immunohistochemistry study, we decided to delete the data on immunohistochemistry in the manuscript to further streamline it. In the current study, although mutations in MSTN lead to a decrease in the cross-sectional area of individual fibers, the number of muscle fibers per unit area were increased, and the final result was an increase in muscle volume and a “double-muscle” phenotype, as well as an increase in the proportion of gluteus medius to carcass (Figure 2K) and the proportion of hind leg meat (Supplementary file 1C). Importantly, there was no significant difference in the proportion of centrally nucleated myofibres between WT and MF^+/-^ sheep (Figure 2-figure supplement 2D), and the elevated expression levels of muscle fiber hypotrophy and muscle atrophy marker genes MTM1, IGF1, SMN1, and GAA are more beneficial for muscle health. Therefore, we support that this is not a muscle fiber hypotrophy. As for the phenotype of muscle fiber hypertrophy demonstrated by most myostatin abrogation studies, we analyzed the possible reasons in the discussion, that is, the effect of MSTN mutation on muscle fiber phenotype may be mutant site-dependent.

Authors then isolated myoblasts from hind limbs of 3-month-old sheep fetuses and cultured in presence of 20% fetal bovine serum before switching to differentiation medium containing 2% horse serum. The cultures showed increased proliferation of Mstn+/-;Fgf5+/- myoblasts as well as downregulation of genes associated with muscle differentiation as well as reduced fusion index. No experiments were performed to assure whether the myostatin and FGF5 pathways were inhibited. No control experiments using supplementation with recombinant proteins and using growth factor depleted culture supplements were performed. As FGF5 and myostatin are secreted factors, evidence is missing whether this led to conditioning of the culture medium. Of note, previous work in mice demonstrated that the double muscle phenotype developed independent of satellite cells activity (Amthor et al. 2009).

We greatly appreciate your valuable suggestions. In addition to detecting the MSTN pathway at the cellular level, we also assayed the expression of MSTN receptors and downstream Smad and Jun families in the gluteus medius, and found that *MSTN*^Del73C^ mutation with *FGF5* knockout led to upregulation of two receptors, while the expression of downstream Smad and Jun families was also inhibited to varying degrees (Figure 4-figure supplement 1A). Considering the possible serum regulation, we also supplemented the data on serum MSTN regulation. Given that the phenotype of MSTN gene editing is mutation site dependent, we directly cultured skeletal muscle satellite cells using serum from WT and MF^+/-^ sheep. We found that serum from MF^+/-^ sheep promoted the proliferation of skeletal muscle satellite cells (Figure 4-figure supplement 1D). *MSTN*^Del73C^ mutation with *FGF5* knockout promoted FOSL1 expression using WT sheep serum (Figure 4-figure supplement 1E), which was similar to the results of FBS culture and HS induction. The serum from MF^+/-^ sheep strongly stimulated FOSL1 expression and the inhibition of MyoD1 (Figure 4-figure supplement 1F). These results indicate that serum regulation cannot be ignored after *MSTN*^Del73C^ mutation with *FGF5* knockout.

Authors then performed RNA seq from Mstn+/-;Fgf5+/- muscle and found a number of differentially expressed genes, but none has been previously reported being involved in the myostatin signaling pathway, so the authors chose to only focus on FOSL1 and associated genes. Authors then demonstrated that Pdpn and Ankrd2 were upregulated during myogenic differentiation, whereas FOPSL1 was downregulated. Moreover, Fosl1 transcription was upregulated in myoblasts and myotubes from Mstn+/-;Fgf5+/- muscle. Authors showed an interaction between Fosl1 and Myod1. Moreover, authors demonstrated that Polsl1 directly binds to the Myod1 promoter. Authors also found decreased p38 MARPK protein levels in proliferating myoblasts from Mstn+/-;Fgf5+/- muscle and increased p38 MARPK in differentiating myotubes.

In the revised version, we have streamlined this section by removing content such as PDPN, AKNRD2, and p38 MAPK, aiming to focus on the MEK-ERK-FOSL1 axis. Meanwhile, we further confirmed the regulatory effect of FOSL1 on MyoD1 by dual luciferase assay.

Furthermore, gain-of-function by overexpressing FOSL1 promoted cell proliferation and inhibited differentiation, and tert-butylhydroquinone, an indirect activator of FOSL1 also inhibited myogenic differentiation. The findings do not support the idea that FOSL1 is not involved, but neither do they strongly support the involvement of FOSL1. The observations made by the authors could be co-incidental and not causative in nature.

We greatly appreciate the valuable suggestions provided by the reviewers, which are of great significance for improving our manuscript. Considering the reviewers’ suggestions, we supplemented the FOSL1 loss-of-function experiments and found that interfering with FOSL1 can inhibit the proliferation and promote differentiation of skeletal muscle satellite cells, which is contrary to the results of overexpression of FOSL1 (Figure 6). Meanwhile, we also used the inhibitor PB98059 to inhibit the ERK pathway to indirectly inhibit the activity of FOSL1, and the results showed that inhibition of FOSL1 activity also promoted myogenic differentiation (Figure 7F-G). These results could further support the important role of FOSL1.

The manuscript by Chen et al. demonstrated successful gene editing in sheep embryos to obtain biallelic mutation of Mstn and FGF5. The resulting double muscle phenotype resulted from fiber hypotrophy and hyperplasia, which contradicts findings in the literature. Chen et al. generated F1 heterozygous offsprings, in which Mstn transcription and translation did not change. Myoblasts from these animals showed increased proliferation and decreased differentiation, which authors interpreted as the underlying cellular mechanism of the double muscle phenotype. However, no work on muscle development in these animals is presented. Important in vitro control experiments are missing. Chen and collaborators found Fosl1 as a differentially expressed gene in Mstn+/-;Fgf5+/- muscle. Fosl1 drives myoblast proliferation and has direct regulatory effect on the Myod1 promoter. The cellular and molecular mechanism of Fosl1 during myogenesis is novel and solid evidence. However, data remain inadequate to conclude whether Fosl1 indeed acts downstream of myostatin.

We greatly appreciate the reviewers for their insightful insights and very constructive suggestions, which were very helpful for further improving our data. In our study, although the mutation in MSTN resulted in a decrease in the cross-sectional area of individual muscle fibers, the number of muscle fibers per unit area increased, which ultimately resulted in an increase in muscle size and the development of a ‘double-muscle’ phenotype. Therefore, we support that this is not a manifestation of muscle fiber dystrophy, and the detection of some marker genes for muscle fiber dystrophy and the proportion of central nucleus of muscle fibers also support this hypothesis (Figure 2-figure supplement 2E-F). In addition, the results such as a reduced cross-sectional area of per muscle fibers in our findings contradict the literature on muscle fiber hypertrophy, which may be due to phenotypic differences caused by mutations at different sites of MSTN, and perhaps may also be species-related. For example, the Belgian blue cattle with a natural mutation in the MSTN gene have an increased number of myofibers and a reduced myofiber cross-sectional area [1], and knockdown of the MSTN gene leads to an increase in the cross-sectional area of muscle fibers in mice, without affecting the number of muscle fibers [2,3], as we further described this in discussion. It should be noted that the possible complementary regulation of FGF5 cannot be ruled out either, but unfortunately, this makes the problem extraordinarily complex. We plan to produce single mutant sheep with segregation of the MSTN and FGF5 genes in subsequent studies and give full consideration to the current problem. Regarding the muscle development of gene-edited animals, due to the limitations of large animal conditions and limited editing individuals, we have not comprehensively evaluated the process of muscle development in vivo to further improve the potential cellular mechanisms of muscle phenotype, except for evaluating the expression of MSTN and FGF5 at the age of 3 months of individual development and the expression of MSTN at 12 months of age (Figure 2-figure supplement 2A). To determine whether FOSL1 indeed acts downstream of MSTN, we supplemented the expression levels of FOSL1 under serum regulation to support our conclusions (Figure 4-figure supplement 1D-F).

[1] Wegner J, Albrecht E, Fiedler I, Teuscher F, Papstein HJ, Ender K. Growth- and breed-related changes of muscle fiber characteristics in cattle[J]. Journal of Animal Science, 2000,78:1485-1496.

[2] Nishi M, Yasue A, Nishimatu S, Nohno T, Yamaoka T, Itakura M, Moriyama K, Ohuchi H, Noji S. A missense mutant myostatin causes hyperplasia without hypertrophy in the mouse muscle[J]. Biochemical and Biophysical Research Communications, 2002,293:247-251.

[3] Zhu X, Hadhazy M, Wehling M, Tidball JG, McNally EM. Dominant negative myostatin produces hypertrophy without hyperplasia in muscle[J]. FEBS Letters, 2000,474:71-75.

As the significant findings are minimal, the amount of text provided, figures and tables are disproportionally excessive. A large number of different molecular techniques are employed to try and decipher the mechanism(s) that result in the observed phenotype = double muscling. The authors focus on the MEK-ERK-FOSL1 pathway an suggest this the key pathway/mechanism resulting in the phenotype observed in MSTNDel273sheep. However, they provide very little solid evidence to support this notion.

Thank you for your review. We have substantially streamlined the manuscript, removed some irrelevant information, and provided all unnecessary figures and tables as supplementary information. Meanwhile, we have added new data to further support that *MSTN*^Del73^ mutation generates a muscle phenotype through the MEK-ERK-FOSL1 pathway.

The manuscript is very long, complicated and difficult to read, given the minimum amount of significant information that is provided. It requires major rewriting to be published. Further, it misses information in material methods, on the generation of animals, on histological techniques and morphometric studies. There is no information provided on the sex of the animals produced and then analyzed. There are also a number of editorial mistakes e.g. the authors refer to tables S1-S4 in the materials and methods and results section, but and there is no table S1-S4 provided.

Thank you for your review. We have greatly streamlined and significantly revised the manuscript. At the same time, we have supplemented detailed information on animal generation, histologic and morphological studies in materials and methods, as well as the information on gene-edited animal production, including gender, age, and so on. Finally, we reviewed the entire manuscript and updated any possible omissions or negligence, such as those oversights like table S1-S4. We have updated table S1-S4 as Supplementary file 2A-D related to the materials, methods, and results section in Supplementary file 2.

**Recommendations for the authors:**
Suggestions to improve the paper (see also public review):- Include the method part of generating the gene edited animals.

We thank the editor and reviewers for pointing out our negligence. We have provided detailed information on the generation method of gene-edited sheep in Materials and Methods, which was produced by injecting MSTN sgRNA, FGF5 sgRNA, and Cas9 mRNA into embryos in different ratios.

- Increase number of Mstn-/-;Fgf5-/- experimental animals allowing for acquisition of statistically relevant data. This is very important as the muscle phenotype of the F1 generation is not obvious. Authors should provide data that the Mstn mutation indeed invalidates myostatin signaling. They should provide data on myostatin protein Mstn transcription as well on myostatin target genes in Mstn-/-;Fgf5-/- sheep.

Many thanks to the eidtor and reviewers for their constructive suggestions. The strategy of using MF^-/-^ sheep to validate the transcription and target gene data of myostatin is indeed the best. However, we only generated one MF^-/-^ sheep, which seriously limits the implementation of such an optimal strategy and may also make statistical analysis based on MF^-/-^ sheep unreliable. Considering these factors, our current study mainly focuses on heterozygous MF^+/-^ sheep. We are planning to generate single gene homozygous mutant sheep for MSTN and FGF5 gene separation in subsequent studies and to give full consideration to the current issue.

- They should also provide data on myostatin target genes in muscles from heterozygous animals.

Thank you for your very informative suggestions. We have quantitatively detected the mRNA expression levels of the receptors and downstream target genes of MSTN in the gluteus medius of heterozygous MF^+/-^ sheep. Compared with WT sheep, the mRNA expression levels of type I receptor (ACVR1) and type II receptor (ACVR2A, ACVR2B) were highly significantly increased in the muscle of MF^+/-^ sheep (Figure 4-figure supplement 1A), there was no significant change in mRNA expression levels in the Smand family (Figure 4-figure supplement 1B), whereas the mRNA expression levels of JunB of Jun family, a downstream target gene of MSTN, were significantly down regulated (Figure 4-figure supplement 1C). These results suggest that the effect of *MSTN*^Del73C^ with *FGF5* knockout may not be limited to MEK-ERK-FOSL1. Again, we would like to thank the editor and reviewers for their constructive suggestions, which provide a new direction for us to further deepen our insight into the mutations of MSTN gene.

- The morphometric results on fiber CSA seem wrong. By looking at the fiber sizes and size bar in Figure 2 H would bring to far higher estimated CSA. There must be a systematic error in using the morphometric algorithm.

Thank you very much for your careful review. There were indeed some errors in morphological analysis of the MF^-/-^ sheep longissimus dorsi and gluteus medius. After checking, we found that the reason why the muscle fiber size was much lower than the data in the previously published sheep report was due to the incorrect use of scale bar. To this end, we re-scanned the tissue slices and used the correct scale bar to re-counted the cross-sectional area of muscle fibers and the number of muscle fiber cells per unit area. In this case, the average cross sectional area of muscle fibers in WT sheep was similar to the previous report.

- The labeling of the ordinate of Fig. 2I is not readable (x1000 µm^2^, or x100 µm^2^?). Authors should make sure that they look at the same muscle part, as fiber sizes can highly vary depending on exact anatomical situation. In small laboratory animals, entire muscle cross sections are usually analyzed to prevent such bias. This may proof difficult in large animals, however, small muscles could easily be identified and cross sections of entire muscles be analyzed. As myostatin KO concerns all skeletal muscles, authors could consider muscle such as FDB or extraocular muscles.

Thank you for your careful review and suggestions. The vertical axis of Figure 2I is in the units of ×1000 μm^2^, and each data point represents the actual measured area of each muscle fiber. Because there are significant differences in muscle fiber size, we visualized the measurement values of all individual muscle fiber areas, and the average value of the scatter plot was used as the average area of all muscle fibers. We did this to provide a more intuitively display the distribution of muscle fiber size.

- The material of methods of muscle histology and morphometric studies must be included.

Thank you for your suggestions. We have supplemented the methods of muscle histology and morphology study, as well as statistical methods for cross-sectional area and quantity of muscle fibers in the material methods.

- In figures, numbers of experimental animals be given throughout, as well as number of technical repeats. The authors need to provide some minimal data on how the genetically engineered sheep were produced, in addition to how many, the sex etc.....and which of these were analyzed to obtain the data. It is impossible to know when reading this manuscript whether data involving, for example gene seq, westerns, microscopic images etc involves one sheep or some compilation of data.

Thank you very much for your constructive suggestions, which is of great guiding significance for improving the quality of our manuscript. We have clearly stated the number of experimental animals and the number of any biological replicates in all figure legends. Meanwhile, we have provided detailed information on the generation method of gene edited sheep in the Materials and Methods, which was produced by injecting MSTN sgRNA, FGF5 sgRNA, and Cas9 mRNA into embryos in different ratios.

- As authors work on Mstn;Fgf5 double KO animals, they should explore whether Fgf5 is expressed in developing sheep muscle, and whether combined KO entails a synergistic effect on muscle development.

We detected the expression of FGF5 in muscle tissue of WT and MF^+/-^ sheep at 3 months of age of individual development, which was significantly reduced compared to WT sheep (Figure 2-figure supplement 2A). We greatly appreciate your very meaningful and valuable comments on the possible synergistic effects of combined knockdown. Due to the limitations of single gene knockout of MSTN and FGF5 in sheep in our current study, especially their homozygous mutants. We will prepare MSTN and FGF5 single gene edited sheep to further explore possible synergistic effects in the following study.

- The authors should address the question of why their mstn mutation causes fiber hypotrophy, whereas most other work reported the opposite. Why would herein generated mutation act differently? Does mutated myostatin gain a different biological effect? Does it bind to different receptors?

Thank you very much for your valuable comment. Regarding the possibility of muscle fiber dystrophy in *MSTN*^Del73C^ mutation with *FGF5* knockout sheep, we have performed a statistical analysis of the proportion of central nucleus of muscle fibers in MF^+/-^ sheep, which can characterize the occurrence of muscle dystrophy in some extent. The results showed that there was no significant difference in the proportion of central nucleus of muscle fibers between WT and MF^+/-^ sheep (Figure 2-figure supplement 2E). At the same time, we also analyzed the mRNA expression levels of genes MTM1, DMD, IGF1, SMN1, and GAA related to muscle fiber dystrophy and muscle atrophy. Although the levels of MTM1, IGF1, SMN1, and GAA were significantly increased (Figure 2-figure supplement 2F), this elevation did not lead to the occurrence of muscle fiber dystrophy and muscle atrophy, but instead, it was beneficial for muscle formation. Therefore, we suggested that this phenomenon produced by *MSTN*^Del73C^ mutation with *FGF5* knockout may not be muscle fiber dystrophy, as *MSTN*^Del73C^ mutation with *FGF5* knockout significantly promoted the proliferation of sheep skeletal muscle satellite cells (Figure 3A-F). More importantly, *MSTN*^Del73C^ mutation with *FGF5* knockout improves the muscle phenotype of sheep, including the "double-muscle" phenotype of the rump (Figure 2A), the proportion of gluteus medius to the carcass (Figure 2K), and the proportion of hind leg meat (Supplementary file 1C). In addition, we analyzed in discussion why the current mutation produces a phenotype different from other work reports, which we suggested that this may be due to different mutation sites. We provided a detailed analysis of this in discussion. It is indeed a very thought-provoking question about whether mutated myostatin acquire different biological effects and whether they bind to different receptors, which we plan to further reveal this in the homozygous MSTN and FGF5 mutant sheep.

- Concerning the in vitro work, authors need to demonstrate whether Mstn and/or FGF5 signaling pathways are altered in myoblasts/myotubes. As both are secreted factors, authors need to show that serum conditioning is changing in myoblast cultures. Authors should perform cultures in which these factors are entirely suppressed and thus signaling pathway shut down. They could use growth factor depleted supplements and/or add myostatin and FGF5 inhibitors to the serum. The need to determine first the individual effect of myostatin and FGF5 and then challenge the combined effect. They also should perform the inverse experiment and supplement cultures with recombinant factors, both as individual approach and combined approach.

We greatly appreciate your valuable suggestions. In addition to detecting the MSTN pathway at the cellular level, we also assayed the expression of MSTN receptors and downstream Smad and Jun families in the gluteus medius, and found that *MSTN*^Del73C^ mutation with *FGF5* knockout led to upregulation of two receptors, while the expression of downstream Smad and Jun families was also inhibited to varying degrees (Figure 4-figure supplement 1A). Considering the possible serum regulation, we also supplemented the data on serum MSTN regulation. Because we have previously tested inhibitors of MSTN and FGF5, but did not observe any effect, we suggest this may be due to the nonspecificity of the inhibitors, as there are no sheep specific MSTN and FGF5 inhibitors. Given that the phenotype of MSTN gene editing is mutation site dependent, we directly cultured skeletal muscle satellite cells using serum from WT and MF^+/-^ sheep. We found that serum from MF^+/-^ sheep promoted the proliferation of skeletal muscle satellite cells (Figure 4-figure supplement 1D). *MSTN*^Del73C^ mutation with *FGF5* knockout promoted FOSL1 expression using WT sheep serum (Figure 4-figure supplement 1E), which was similar to the results of FBS culture and HS induction. The serum from MF^+/-^ sheep strongly stimulated FOSL1 expression and the inhibition of MyoD1 (Figure 4-figure supplement 1F). These results indicate that serum regulation cannot be ignored after *MSTN*^Del73C^ mutation with *FGF5* knockout.

- With above suggested additional experiments, authors would also be able to demonstrate, whether Fosl1 is indeed triggered in response to myostatin and/or FGF5 signaling.

To determine whether FOSL1 indeed acts downstream of MSTN, we supplemented the expression levels of FOSL1 under serum regulation to support our conclusions. We found that the serum from MF^+/-^ sheep strongly stimulated FOSL1 expression and the inhibition of MyoD1 (Figure 4—figure supplement 1F).

- Authors used t-test despite in several tests despite low sample number, which violates as such the assumption of equal variance. Non-parametric tests should be used in this case.

Thank you very much for your valuable comments. We apologize for the previous incorrect use of statistical methods. In the revised version, we have re-analyzed all data. Before performing student’s t-test, we first evaluated the assumptions of normal distribution and equal variance. Two-tailed student’s t-tests were used only for data that conformed to normal distribution and homogeneity of variance, otherwise corrected Welch's t-tests were performed.

- Authors should state in the legends which statistical test was used.

Thank you for your suggestion. We have clearly stated the statistical testing method used in all figure legends, which is indeed necessary and important.

In general, this manuscript should be dramatically scaled back in terms of content, eliminating unnecessary text, figures and tables that do not play a significant role in the findings that were significant. There is some interesting information and data here that can add to the overall base of knowledge surrounding the production of genetically engineered livestock in which myostatin has been targeted for mutation. However, the authors need to focus on their findings that were significant and strongly supported by the data and statistical analysis. Some discussion of findings that support their ideas/hypothesis, but are not statistically significant is fine. But it should not make up the majority of the manuscript which is the case here.

Thank you for your valuable suggestions, which are essential for improving the quality of our manuscript. We have greatly streamlined and significantly revised the manuscript, removed unnecessary text, figures, and tables.